# Bayesian Deep Ensembles via the Neural Tangent Kernel

**Bobby He**
Department of Statistics
University of Oxford
bobby.he@stats.ox.ac.uk

**Balaji Lakshminarayanan**
Google Research
Brain team
balajiln@google.com

**Yee Whye Teh**
Department of Statistics
University of Oxford
y.w.teh@stats.ox.ac.uk

## Abstract

We explore the link between deep ensembles and Gaussian processes (GPs) through the lens of the Neural Tangent Kernel (NTK): a recent development in understanding the training dynamics of wide neural networks (NNs). Previous work has shown that even in the infinite width limit, when NNs become GPs, there is no GP posterior interpretation to a deep ensemble trained with squared error loss. We introduce a simple modification to standard deep ensembles training, through addition of a computationally-tractable, randomised and untrainable function to each ensemble member, that enables a posterior interpretation in the infinite width limit. When ensembled together, our trained NNs give an approximation to a posterior predictive distribution, and we prove that our Bayesian deep ensembles make more conservative predictions than standard deep ensembles in the infinite width limit. Finally, using finite width NNs we demonstrate that our Bayesian deep ensembles faithfully emulate the analytic posterior predictive when available, and can outperform standard deep ensembles in various out-of-distribution settings, for both regression and classification tasks.

## 1 Introduction

Consider a training dataset $\mathcal{D}$ consisting of $N$ i.i.d. data points $\mathcal{D} = \{\mathcal{X}, \mathcal{Y}\} = \{(\boldsymbol{x}_n, y_n)\}_{n=1}^N$, with $\boldsymbol{x} \in \mathbb{R}^d$ representing $d$-dimensional features and $y$ representing $C$-dimensional targets. Given input features $\boldsymbol{x}$ and parameters $\boldsymbol{\theta} \in \mathbb{R}^p$ we use the output, $f(\boldsymbol{x}, \boldsymbol{\theta}) \in \mathbb{R}^C$, of a neural network (NN) to model the predictive distribution $p(y|\boldsymbol{x}, \boldsymbol{\theta})$ over the targets. For univariate regression tasks, $p(y|\boldsymbol{x}, \boldsymbol{\theta})$ will be Gaussian: $-\log p(y|\boldsymbol{x}, \boldsymbol{\theta})$ is the squared error $\frac{1}{2\sigma^2}(y - f(\boldsymbol{x}, \boldsymbol{\theta}))^2$ up to additive constant, for fixed observation noise $\sigma^2 \in \mathbb{R}_+$. For classification tasks, $p(y|\boldsymbol{x}, \boldsymbol{\theta})$ will be a Categorical distribution.

Given a prior distribution $p(\boldsymbol{\theta})$ over the parameters, we can define the posterior over $\boldsymbol{\theta}$, $p(\boldsymbol{\theta}|\mathcal{D})$, using Bayes' rule and subsequently the *posterior predictive* distribution at a test point $(\boldsymbol{x}^*, y^*)$:

$$p(y^*|\boldsymbol{x}^*, \mathcal{D}) = \int p(y^*|\boldsymbol{x}^*, \boldsymbol{\theta}) p(\boldsymbol{\theta}|\mathcal{D}) \, d\boldsymbol{\theta} \tag{1}$$

The posterior predictive is appealing as it represents a marginalisation over $\boldsymbol{\theta}$ weighted by posterior probabilities, and has been shown to be optimal for minimising predictive risk under a well-specified model [1]. However, one issue with the posterior predictive for NNs is that it is computationally intensive to calculate the posterior $p(\boldsymbol{\theta}|\mathcal{D})$ exactly. Several approximations to $p(\boldsymbol{\theta}|\mathcal{D})$ have been introduced for *Bayesian neural networks* (BNNs) including: Laplace approximation [2]; Markov chain Monte Carlo [3, 4]; variational inference [5–9]; and Monte-Carlo dropout [10].

Despite the recent interest in BNNs, it has been shown empirically that deep ensembles [11], which lack a principled Bayesian justification, outperform existing BNNs in terms of uncertainty quantification and out-of-distribution robustness, cf. [12]. Deep ensembles independently initialise and train individual NNs (referred to herein as *baselearners*) on the negative log-likelihood loss

$\mathcal{L}(\boldsymbol{\theta}) = \sum_{n=1}^{N} \ell(y_n, f(\boldsymbol{x}_n, \boldsymbol{\theta}))$ with $\ell(y, f(\boldsymbol{x}, \boldsymbol{\theta})) = -\log p(y|\boldsymbol{x}, \boldsymbol{\theta})$, before aggregating predictions. Understanding the success of deep ensembles, particularly in relation to Bayesian inference, is a key question in the uncertainty quantification and Bayesian deep learning communities at present: Fort et al. [13] suggested that the empirical performance of deep ensembles is explained by their ability to explore different functional modes, while Wilson and Izmailov [14] argued that deep ensembles are actually approximating the posterior predictive.

In this work, we will relate deep ensembles to Bayesian inference, using recent developments connecting GPs and wide NNs, both before [15–21] and after [22, 23] training. Using these insights, we devise a modification to standard NN training that yields an exact posterior sample for $f(\cdot, \boldsymbol{\theta})$ in the infinite width limit. As a result, when ensembled together our modified baselearners give a posterior predictive approximation, and can thus be viewed as a *Bayesian deep ensemble*.

One concept that is related to our methods concerns ensembles trained with *Randomised Priors* to give an approximate posterior interpretation, which we will use when modelling observation noise in regression tasks. The idea behind randomised priors is that, under certain conditions, regularising baselearner NNs towards independently drawn "priors" during training produces exact posterior samples for $f(\cdot, \boldsymbol{\theta})$. Randomised priors recently appeared in machine learning applied to reinforcement learning [24] and uncertainty quantification [25, 26], like this work. To the best of our knowledge, related ideas first appeared in astrophysics where they were applied to Gaussian random fields [27]. However, one such condition for posterior exactness with randomised priors is that the model $f(\boldsymbol{x}, \boldsymbol{\theta})$ is linear in $\boldsymbol{\theta}$. This is not true in general for NNs, but has been shown to hold for wide NNs local to their parameter initialisation, in a recent line of work. In order to introduce our methods, we will first review this line of work, known as the *Neural Tangent Kernel* (NTK) [22].

## 2 NTK Background

Wide NNs, and their relation to GPs, have been a fruitful area recently for the theoretical study of NNs: we review only the most salient developments to this work, due to limited space.

First introduced by Jacot et al. [22], the *empirical NTK* of $f(\cdot, \boldsymbol{\theta}_t)$ is, for inputs $\boldsymbol{x}, \boldsymbol{x}'$, the kernel:

$$\hat{\Theta}_t(\boldsymbol{x}, \boldsymbol{x}') = \langle \nabla_{\boldsymbol{\theta}} f(\boldsymbol{x}, \boldsymbol{\theta}_t), \nabla_{\boldsymbol{\theta}} f(\boldsymbol{x}', \boldsymbol{\theta}_t) \rangle \tag{2}$$

and describes the functional gradient of a NN in terms of the current loss incurred on the training set. Note that $\boldsymbol{\theta}_t$ depends on a random initialisation $\boldsymbol{\theta}_0$, thus the empirical NTK is random for all $t > 0$.

Jacot et al. [22] showed that for an MLP under a so-called NTK parameterisation, detailed in Appendix A, the empirical NTK converges in probability to a deterministic limit $\Theta$, that stays constant during gradient training, as the hidden layer widths of the NN go to infinity sequentially. Later, Yang [28, 29] extended the NTK convergence result to convergence almost surely, which is proven rigorously for a variety of architectures and for widths (or channels in Convolutional NNs) of hidden layers going to infinity in unison. This limiting positive-definite (p.d.) kernel $\Theta$, known as the NTK, depends only on certain NN architecture choices, including: activation, depth and variances for weight and bias parameters. Note that the NTK parameterisation can be thought of as akin to training under standard parameterisation with a learning rate that is inversely proportional to the width of the NN, which has been shown to be the largest scale for stable learning rates in wide NNs [30–32].

Lee et al. [23] built on the results of Jacot et al. [22], and studied the *linearised regime* of an NN. Specifically, if we denote as $f_t(\boldsymbol{x}) = f(\boldsymbol{x}, \boldsymbol{\theta}_t)$ the network function at time $t$, we can define the first order Taylor expansion of the network function around randomly initialised parameters $\boldsymbol{\theta}_0$ to be:

$$f_t^{\text{lin}}(\boldsymbol{x}) = f_0(\boldsymbol{x}) + \nabla_{\boldsymbol{\theta}} f(\boldsymbol{x}, \boldsymbol{\theta}_0) \Delta \boldsymbol{\theta}_t \tag{3}$$

where $\Delta \boldsymbol{\theta}_t = \boldsymbol{\theta}_t - \boldsymbol{\theta}_0$ and $f_0 = f(\cdot, \boldsymbol{\theta}_0)$ is the randomly initialised NN function.

For notational clarity, whenever we evaluate a function at an arbitrary input set $\mathcal{X}'$ instead of a single point $\boldsymbol{x}'$, we suppose the function is vectorised. For example, $f_t(\mathcal{X}) \in \mathbb{R}^{NC}$ denotes the concatenated NN outputs on training set $\mathcal{X}$, whereas $\nabla_{\boldsymbol{\theta}} f_t(\mathcal{X}) = \nabla_{\boldsymbol{\theta}} f(\mathcal{X}, \boldsymbol{\theta}_t) \in \mathbb{R}^{NC \times p}$. In the interest of space, we will also sometimes use subscripts to signify kernel inputs, so for instance $\Theta_{\boldsymbol{x}'\mathcal{X}} = \Theta(\boldsymbol{x}', \mathcal{X}) \in \mathbb{R}^{C \times NC}$ and $\Theta_{\mathcal{X}\mathcal{X}} = \Theta(\mathcal{X}, \mathcal{X}) \in \mathbb{R}^{NC \times NC}$ throughout this work.

The results of Lee et al. [23] showed that in the infinite width limit, with NTK parameterisation and gradient flow under squared error loss, $f_t^{\text{lin}}(\boldsymbol{x})$ and $f_t(\boldsymbol{x})$ are equal for any $t \geq 0$, for a shared random

initialisation $\boldsymbol{\theta}_0$. In particular, for the linearised network it can be shown, that as $t\to\infty$:

$$f_\infty^{\text{lin}}(\boldsymbol{x}) = f_0(\boldsymbol{x}) - \hat{\Theta}_0(\boldsymbol{x},\mathcal{X})\hat{\Theta}_0(\mathcal{X},\mathcal{X})^{-1}(f_0(\mathcal{X}) - \mathcal{Y}) \tag{4}$$

and thus as the hidden layer widths converge to infinity we have that:

$$f_\infty^{\text{lin}}(\boldsymbol{x}) = f_\infty(\boldsymbol{x}) = f_0(\boldsymbol{x}) - \Theta(\boldsymbol{x},\mathcal{X})\Theta(\mathcal{X},\mathcal{X})^{-1}(f_0(\mathcal{X}) - \mathcal{Y}) \tag{5}$$

We can replace $\Theta(\mathcal{X},\mathcal{X})^{-1}$ with the generalised inverse when invertibility is an issue. However, this will not be a main concern of this work, as our methods will add regularisation that corresponds to modelling observation/output noise, which both ensures invertibility and alleviates any potential convergence issues due to fast decay of the NTK eigenspectrum [33].

From Eq. (5) we see that, conditional on the training data $\{\mathcal{X},\mathcal{Y}\}$, we can decompose $f_\infty$ into $f_\infty(\boldsymbol{x}) = \mu(\boldsymbol{x})+\gamma(\boldsymbol{x})$ where $\mu(\boldsymbol{x}) = \Theta(\boldsymbol{x},\mathcal{X})\Theta(\mathcal{X},\mathcal{X})^{-1}\mathcal{Y}$ is a deterministic mean and $\gamma(\boldsymbol{x}) = f_0(\boldsymbol{x})-\Theta(\boldsymbol{x},\mathcal{X})\Theta(\mathcal{X},\mathcal{X})^{-1}f_0(\mathcal{X})$ captures predictive uncertainty, due to the randomness of $f_0$. Now, if we suppose that, at initialisation, $f_0 \overset{d}{\sim} \mathcal{GP}(0,k)$ for an arbitrary kernel $k:\mathbb{R}^d\times\mathbb{R}^d \to \mathbb{R}^{C\times C}$, then we have $f_\infty(\cdot) \overset{d}{\sim} \mathcal{GP}(\mu(\boldsymbol{x}),\Sigma(\boldsymbol{x},\boldsymbol{x}'))$ for two inputs $\boldsymbol{x}, \boldsymbol{x}'$, where:[1]

$$\Sigma(\boldsymbol{x},\boldsymbol{x}') = k_{\boldsymbol{x}\boldsymbol{x}'} + \Theta_{\boldsymbol{x}\mathcal{X}}\Theta_{\mathcal{X}\mathcal{X}}^{-1}k_{\mathcal{X}\mathcal{X}}\Theta_{\mathcal{X}\mathcal{X}}^{-1}\Theta_{\mathcal{X}\boldsymbol{x}} - \left(\Theta_{\boldsymbol{x}\mathcal{X}}\Theta_{\mathcal{X}\mathcal{X}}^{-1}k_{\mathcal{X}\boldsymbol{x}'} + h.c.\right) \tag{6}$$

For a generic kernel $k$, Lee et al. [23] observed that this limiting distribution for $f_\infty$ does not have a posterior GP interpretation unless $k$ and $\Theta$ are multiples of each other.

As mentioned in Section 1, previous work [15–21] has shown that there is a distinct but closely related kernel $\mathcal{K}$, known as the *Neural Network Gaussian Process* (NNGP) kernel, such that $f_0 \overset{d}{\to} \mathcal{GP}(0,\mathcal{K})$ at initialisation in the infinite width limit and $\mathcal{K} \neq \Theta$. Thus Eq. (6) with $k=\mathcal{K}$ tells us that, for wide NNs under squared error loss, there is no Bayesian posterior interpretation to a trained NN, nor is there an interpretation to a trained deep ensemble as a Bayesian posterior predictive approximation.

## 3   Proposed modification to obtain posterior samples in infinite width limit

Lee et al. [23] noted that one way to obtain a posterior interpretation to $f_\infty$ is by randomly initialising $f_0$ but only training the parameters in the final linear readout layer, as the contribution to the NTK $\Theta$ from the parameters in final hidden layer is exactly the NNGP kernel $\mathcal{K}$.[2] $f_\infty$ is then a sample from the GP posterior with prior kernel NNGP, $\mathcal{K}$, and noiseless observations in the infinite width limit i.e. $f_\infty(\mathcal{X}') \overset{d}{\sim} \mathcal{N}(\mathcal{K}_{\mathcal{X}'\mathcal{X}}\mathcal{K}_{\mathcal{X}\mathcal{X}}^{-1}\mathcal{Y},\ \mathcal{K}_{\mathcal{X}'\mathcal{X}'} - \mathcal{K}_{\mathcal{X}'\mathcal{X}}\mathcal{K}_{\mathcal{X}\mathcal{X}}^{-1}\mathcal{K}_{\mathcal{X}\mathcal{X}'})$. This is an example of the "sample-then-optimise" procedure of Matthews et al. [34], but, by only training the final layer this procedure limits the earlier layers of an NN solely to be random feature extractors.

We now introduce our modification to standard training that trains all layers of a finite width NN and obtains an exact posterior interpretation in the infinite width limit with NTK parameterisation and squared error loss. For notational purposes, let us suppose $\boldsymbol{\theta}=\texttt{concat}(\{\boldsymbol{\theta}^{\leq L},\boldsymbol{\theta}^{L+1}\})$ with $\boldsymbol{\theta}^{\leq L}\in\mathbb{R}^{p-p_{L+1}}$ denoting $L$ hidden layers, and $\boldsymbol{\theta}^{L+1}\in\mathbb{R}^{p_{L+1}}$ denoting final readout layer $L+1$. Moreover, define $\Theta^{\leq L}=\Theta-\mathcal{K}$ to be the p.d. kernel corresponding to contributions to the NTK from all parameters before the final layer, and $\hat{\Theta}_t^{\leq L}$ to be the empirical counterpart depending on $\boldsymbol{\theta}_t$. To motivate our modification, we reinterpret $f_t^{\text{lin}}$ in Eq. (3) by splitting terms related to $\mathcal{K}$ and $\Theta^{\leq L}$:

$$f_t^{\text{lin}}(\boldsymbol{x}) = \underbrace{f_0(\boldsymbol{x}) + \nabla_{\boldsymbol{\theta}^{L+1}}f(\boldsymbol{x},\boldsymbol{\theta}_0)\Delta\boldsymbol{\theta}_t^{L+1}}_{\mathcal{K}} + \underbrace{\boldsymbol{0}_C + \nabla_{\boldsymbol{\theta}^{\leq L}}f(\boldsymbol{x},\boldsymbol{\theta}_0)\Delta\boldsymbol{\theta}_t^{\leq L}}_{\Theta-\mathcal{K}} \tag{7}$$

where $\boldsymbol{0}_C\in\mathbb{R}^C$ is the zero vector. As seen in Eq. (7), the distribution of $f_0^{\text{lin}}(\boldsymbol{x})=f_0(\boldsymbol{x})$ lacks extra variance, $\Theta^{\leq L}(\boldsymbol{x},\boldsymbol{x})$, that accounts for contributions to the NTK $\Theta$ from all parameters $\boldsymbol{\theta}^{\leq L}$ before the final layer. This is precisely why no Bayesian intepretation exists for a standard trained wide NN, as in Eq. (6) with $k=\mathcal{K}$. The motivation behind our modification is now very simple: we propose to manually add in this missing variance. Our modified NNs, $\tilde{f}(\cdot,\boldsymbol{\theta})$, will then have trained distribution:

$$\tilde{f}_\infty(\mathcal{X}') \overset{d}{\sim} \mathcal{N}(\Theta_{\mathcal{X}'\mathcal{X}}\Theta_{\mathcal{X}\mathcal{X}}^{-1}\mathcal{Y},\ \Theta_{\mathcal{X}'\mathcal{X}'} - \Theta_{\mathcal{X}'\mathcal{X}}\Theta_{\mathcal{X}\mathcal{X}}^{-1}\Theta_{\mathcal{X}\mathcal{X}'}) \tag{8}$$

  For example: $\Theta_{\boldsymbol{x}\mathcal{X}}\Theta_{\mathcal{X}\mathcal{X}}^{-1}k_{\mathcal{X}\boldsymbol{x}'} + h.c. = \Theta_{\boldsymbol{x}\mathcal{X}}\Theta_{\mathcal{X}\mathcal{X}}^{-1}k_{\mathcal{X}\boldsymbol{x}'} + \Theta_{\boldsymbol{x}'\mathcal{X}}\Theta_{\mathcal{X}\mathcal{X}}^{-1}k_{\mathcal{X}\boldsymbol{x}}$

[2]Up to a multiple of last layer width in standard parameterisation.

on a test set $\mathcal{X}'$, in the infinite width limit. Note that Eq. (8) is the GP posterior using prior kernel $\Theta$ and noiseless observations $\tilde{f}_\infty(\mathcal{X}) = \mathcal{Y}$, which we will refer to as the *NTKGP* posterior predictive. We construct $\tilde{f}$ by sampling a random and untrainable function $\delta(\cdot)$ that is added to the standard forward pass $f(\cdot, \boldsymbol{\theta}_t)$, defining an augmented forward pass:

$$\tilde{f}(\cdot, \boldsymbol{\theta}_t) = f(\cdot, \boldsymbol{\theta}_t) + \delta(\cdot) \tag{9}$$

Given a parameter initialisation scheme $\texttt{init}(\cdot)$ and initial parameters $\boldsymbol{\theta}_0 \overset{d}{\sim} \texttt{init}(\cdot)$, our chosen formulation for $\delta(\cdot)$ is as follows: 1) sample $\tilde{\boldsymbol{\theta}} \overset{d}{\sim} \texttt{init}(\cdot)$ independently of $\boldsymbol{\theta}_0$; 2) denote $\tilde{\boldsymbol{\theta}} = \texttt{concat}(\{\tilde{\boldsymbol{\theta}}^{\leq L}, \tilde{\boldsymbol{\theta}}^{L+1}\})$; and 3) define $\boldsymbol{\theta}^* = \texttt{concat}(\{\tilde{\boldsymbol{\theta}}^{\leq L}, \mathbf{0}_{p_{L+1}}\})$. In words, we set the parameters in the final layer of an independently sampled $\tilde{\boldsymbol{\theta}}$ to zero to obtain $\boldsymbol{\theta}^*$. Now, we define:

$$\delta(\boldsymbol{x}) = \nabla_{\boldsymbol{\theta}} f(\boldsymbol{x}, \boldsymbol{\theta}_0) \boldsymbol{\theta}^* \tag{10}$$

There are a few important details to note about $\delta(\cdot)$ as defined in Eq. (10). First, $\delta(\cdot)$ has the same distribution in both NTK and standard parameterisations,[3] and also $\delta(\cdot) \mid \boldsymbol{\theta}_0 \overset{d}{\sim} \mathcal{GP}(0, \hat{\Theta}_0^{\leq L})$ in the NTK parameterisation.[4] Moreover, Eq. (10) can be viewed as a single Jacobian-vector product (JVP), which packages that offer forward-mode autodifferentiation (AD), such as JAX [35], are efficient at computing for finite NNs. It is worth noting that our modification adds only negligible computational and memory requirements on top of standard deep ensembles [11]: a more nuanced comparison can be found in Appendix G. Alternative constructions of $\tilde{f}$ are presented in Appendix C.

To ascertain whether a trained $\tilde{f}_\infty$ constructed via Eqs. (9, 10) returns a sample from the GP posterior Eq. (8) for wide NNs, the following proposition, which we prove in Appendix B.1, will be useful:

**Proposition 1.** $\delta(\cdot) \overset{d}{\to} \mathcal{GP}(0, \Theta^{\leq L})$ *and is independent of $f_0(\cdot)$ in the infinite width limit. Thus,* $\tilde{f}_0(\cdot) = f_0(\cdot) + \delta(\cdot) \overset{d}{\to} \mathcal{GP}(0, \Theta)$.

Using Proposition 1, we now consider the linearisation of $\tilde{f}_t(\cdot)$, noting that $\nabla_{\boldsymbol{\theta}} \tilde{f}_0(\cdot) = \nabla_{\boldsymbol{\theta}} f_0(\cdot)$:

$$\tilde{f}_t^{\text{lin}}(\boldsymbol{x}) = \underbrace{f_0(\boldsymbol{x}) + \nabla_{\boldsymbol{\theta}^{L+1}} f(\boldsymbol{x}, \boldsymbol{\theta}_0) \Delta\boldsymbol{\theta}_t^{L+1}}_{\mathcal{K}} + \underbrace{\delta(\boldsymbol{x}) + \nabla_{\boldsymbol{\theta}^{\leq L}} f(\boldsymbol{x}, \boldsymbol{\theta}_0) \Delta\boldsymbol{\theta}_t^{\leq L}}_{\Theta - \mathcal{K}} \tag{11}$$

The fact that $\nabla_{\boldsymbol{\theta}} \tilde{f}_t^{\text{lin}}(\cdot) = \nabla_{\boldsymbol{\theta}} f_0(\cdot)$ is crucial in Eq. (11), as this initial Jacobian is the feature map of the linearised NN regime from Lee et al. [23]. As per Proposition 1 and Eq. (11), we see that $\delta(\boldsymbol{x})$ adds the extra randomness missing from $f_0^{\text{lin}}(\boldsymbol{x})$ in Eq. (7), and reinitialises $\tilde{f}_0$ as a sample from $\mathcal{GP}(0, \mathcal{K})$ to $\mathcal{GP}(0, \Theta)$ for wide NNs. This means we can set $k = \Theta$ in Eq. (6) and deduce:

**Corollary 1.** $\tilde{f}_\infty(\mathcal{X}') \overset{d}{\sim} \mathcal{N}(\Theta_{\mathcal{X}'\mathcal{X}} \Theta_{\mathcal{X}\mathcal{X}}^{-1} \mathcal{Y}, \ \Theta_{\mathcal{X}'\mathcal{X}'} - \Theta_{\mathcal{X}'\mathcal{X}} \Theta_{\mathcal{X}\mathcal{X}}^{-1} \Theta_{\mathcal{X}\mathcal{X}'})$, *and hence a trained $\tilde{f}_\infty$ returns a sample from the posterior NTKGP in the infinite width limit.*

To summarise: we define our new NN forward pass to give $\tilde{f}_t(\boldsymbol{x}) = f_t(\boldsymbol{x}) + \delta(\boldsymbol{x})$ for standard forward pass $f_t(\boldsymbol{x})$, and an untrainable $\delta(\boldsymbol{x})$ defined as in Eq. (10). As given by Corollary 1, independently trained baselearners $\tilde{f}_\infty$ can then be ensembled to approximate the NTKGP posterior predictive.

We will call $\tilde{f}_\infty$ trained in this section an NTKGP baselearner, regardless of parameterisation or width. We are aware that the name NTK-GP has been used previously to refer to Eq. (6) with NNGP kernel $\mathcal{K}$, which is what standard training under squared error with a wide NN yields. However, we believe GPs in machine learning are synonymous with probabilistic inference [36], which Eq. (6) has no connection to in general, so we feel the name NTKGP is more appropriate for our methods.

## 3.1 Modelling observation noise

So far, we have used squared loss $\ell(y, \tilde{f}(\boldsymbol{x}, \boldsymbol{\theta})) = \frac{1}{2\sigma^2}(y - \tilde{f}(\boldsymbol{x}, \boldsymbol{\theta}))^2$ for $\sigma^2{=}1$, and seen how our NTKGP training scheme for $\tilde{f}$ gives a Bayesian interpretation to trained networks when we assume noiseless observations. Lemma 3 of Osband et al. [24] shows us how to draw a posterior sample for linear $\tilde{f}$ if we wish to model Gaussian observation noise $y \stackrel{d}{\sim} \mathcal{N}(\tilde{f}(\boldsymbol{x}, \boldsymbol{\theta}), \sigma^2)$ for $\sigma^2{>}0$: by adding i.i.d. noise to targets $y'_n \stackrel{d}{\sim} \mathcal{N}(y_n, \sigma^2)$ and regularising $\mathcal{L}(\boldsymbol{\theta})$ with a weighted $L^2$ term, either $\|\boldsymbol{\theta}\|_\Lambda^2$ or $\|\boldsymbol{\theta} - \boldsymbol{\theta}_0\|_\Lambda^2$, depending on if you regularise in function space or parameter space. The weighting $\Lambda$ is detailed in Appendix D. These methods were introduced by Osband et al. [24] for the application of Q-learning in deep reinforcement learning, and are known as *Randomised Prior parameter* (RP-param) and *Randomised Prior function* (RP-fn) respectively. The randomised prior (RP) methods were motivated by a Bayesian linear regression approximation of the NN, but they do not take into account the difference between the NNGP and the NTK. Our NTKGP methods can be viewed as a way to fix this for both the parameter space or function space methods, which we will name NTKGP-param and NTKGP-fn respectively. Similar regularisation ideas were explored in connection to the NTK by Hu et al. [37], when the NN function is initialised from the origin, akin to kernel ridge regression.

## 3.2 Comparison of predictive distributions in infinite width

Having introduced the different ensemble training methods considered in this paper: NNGP; deep ensembles; randomised prior; and NTKGP, we will now compare their predictive distributions in the infinite width limit with squared error loss. Table 1 displays these limiting distributions, $f_\infty(\cdot) \stackrel{d}{\sim} \mathcal{GP}(\mu, \Sigma)$, and should be viewed as an extension to Equation (16) of Lee et al. [23]. In

Table 1: Predictive distributions of wide ensembles for various training methods. *std* denotes standard training with $f(\boldsymbol{x}, \boldsymbol{\theta})$, and *ours* denotes training using our additive $\delta(\boldsymbol{x})$ to make $\tilde{f}(\boldsymbol{x}, \boldsymbol{\theta})$.

| Method | Layers trained | Output Noise | $\mu(\boldsymbol{x})$ | $\Sigma(\boldsymbol{x}, \boldsymbol{x}')$ |
|---|---|---|---|---|
| NNGP | Final | $\sigma^2 \geq 0$ | $\mathcal{K}_{\boldsymbol{x}\mathcal{X}}(\mathcal{K}_{\mathcal{X}\mathcal{X}} + \sigma^2 I)^{-1}\mathcal{Y}$ | $\mathcal{K}_{\boldsymbol{x}\boldsymbol{x}'} - \mathcal{K}_{\boldsymbol{x}\mathcal{X}}(\mathcal{K}_{\mathcal{X}\mathcal{X}} + \sigma^2 I)^{-1}\mathcal{K}_{\mathcal{X}\boldsymbol{x}'}$ |
| Deep Ensembles | All (*std*) | $\sigma^2 = 0$ | $\Theta_{\boldsymbol{x}\mathcal{X}}\Theta_{\mathcal{X}\mathcal{X}}^{-1}\mathcal{Y}$ | $\mathcal{K}_{\boldsymbol{x}\boldsymbol{x}'} - (\Theta_{\boldsymbol{x}\mathcal{X}}\Theta_{\mathcal{X}\mathcal{X}}^{-1}\mathcal{K}_{\mathcal{X}\boldsymbol{x}'} + h.c.)$ $\Theta_{\boldsymbol{x}\mathcal{X}}\Theta_{\mathcal{X}\mathcal{X}}^{-1}\mathcal{K}_{\mathcal{X}\mathcal{X}}\Theta_{\mathcal{X}\mathcal{X}}^{-1}\Theta_{\mathcal{X}\boldsymbol{x}'}$ |
| Randomised Prior | All (*std*) | $\sigma^2 > 0$ | $\Theta_{\boldsymbol{x}\mathcal{X}}(\Theta_{\mathcal{X}\mathcal{X}} + \sigma^2 I)^{-1}\mathcal{Y}$ | $\mathcal{K}_{\boldsymbol{x}\boldsymbol{x}'} - (\Theta_{\boldsymbol{x}\mathcal{X}}(\Theta_{\mathcal{X}\mathcal{X}} + \sigma^2 I)^{-1}\mathcal{K}_{\mathcal{X}\boldsymbol{x}'} + h.c.)$ $+ \Theta_{\boldsymbol{x}\mathcal{X}}(\Theta_{\mathcal{X}\mathcal{X}} + \sigma^2 I)^{-1}(\mathcal{K}_{\mathcal{X}\mathcal{X}} + \sigma^2 I)(\Theta_{\mathcal{X}\mathcal{X}} + \sigma^2 I)^{-1}\Theta_{\mathcal{X}\boldsymbol{x}'}$ |
| NTKGP | All (*ours*) | $\sigma^2 \geq 0$ | $\Theta_{\boldsymbol{x}\mathcal{X}}(\Theta_{\mathcal{X}\mathcal{X}} + \sigma^2 I)^{-1}\mathcal{Y}$ | $\Theta_{\boldsymbol{x}\boldsymbol{x}'} - \Theta_{\boldsymbol{x}\mathcal{X}}(\Theta_{\mathcal{X}\mathcal{X}} + \sigma^2 I)^{-1}\Theta_{\mathcal{X}\boldsymbol{x}'}$ |

order to parse Table 1, let us denote $\mu_{\text{NNGP}}, \mu_{\text{DE}}, \mu_{\text{RP}}, \mu_{\text{NTKGP}}$ and $\Sigma_{\text{NNGP}}, \Sigma_{\text{DE}}, \Sigma_{\text{RP}}, \Sigma_{\text{NTKGP}}$ to be the entries in the $\mu(\boldsymbol{x})$ and $\Sigma(\boldsymbol{x}, \boldsymbol{x}')$ columns of Table 1 respectively, read from top to bottom. We see that $\mu_{\text{DE}}(\boldsymbol{x}) = \mu_{\text{NTKGP}}(\boldsymbol{x})$ if $\sigma^2{=}0$, and $\mu_{\text{RP}}(\boldsymbol{x}) = \mu_{\text{NTKGP}}(\boldsymbol{x})$ if $\sigma^2{>}0$. In words: the predictive mean of a trained ensemble is the same when training all layers, both with standard training and our NTKGP training. This holds because both $f_0$ and $\tilde{f}_0$ are zero mean. It is also possible to compare the predictive covariances as the following proposition, proven in Appendix B.2, shows:

**Proposition 2.** *For $\sigma^2{=}0$, $\Sigma_{NTKGP} \succeq \Sigma_{DE} \succeq \Sigma_{NNGP}$. Similarly, for $\sigma^2{>}0$, $\Sigma_{NTKGP} \succeq \Sigma_{RP} \succeq \Sigma_{NNGP}$.*

Here, when we write $k_1 \succeq k_2$ for p.d. kernels $k_1, k_2$, we mean that $k_1 - k_2$ is also a p.d. kernel. One consequence of Proposition 2 is that the predictive distribution of an ensemble of NNs trained via our NTKGP methods is always more conservative than a standard deep ensemble, in the linearised NN regime, when the ensemble size $K{\to}\infty$. It is not possible to say in general when this will be beneficial, because in practice our models will always be misspecified. However, Proposition 2 suggests that in situations where we suspect standard deep ensembles might be overconfident, such as in situations where we expect some dataset shift at test time, our methods should hold an advantage. Note, wide randomised prior ensembles ($\sigma > 0$) were also theoretically shown to make more conservative predictions than corresponding NNGP posteriors in Ciosek et al. [26], albeit without the connection to the NTK.

### 3.3 Modelling heteroscedasticity

Following Lakshminarayanan et al. [11], if we wish to model heteroscedasticity in a univariate regression setting such that each training point, $(\boldsymbol{x}_n, y_n)$, has an individual observation noise $\sigma^2(\boldsymbol{x}_n)$ then we use the heteroscedastic Gaussian NLL loss (up to additive constant):

$$\ell(y'_n, \tilde{f}(\boldsymbol{x}_n, \boldsymbol{\theta})) = \frac{(y'_n - \tilde{f}(\boldsymbol{x}_n, \boldsymbol{\theta}))^2}{2\sigma^2(\boldsymbol{x}_n)} + \frac{\log \sigma^2(\boldsymbol{x}_n)}{2} \tag{12}$$

where $y'_n = y_n + \sigma(\boldsymbol{x}_n)\epsilon_n$ and $\epsilon_n \overset{\text{i.i.d.}}{\sim} \mathcal{N}(0, 1)$. It is easy to see that for fixed $\sigma^2(\boldsymbol{x}_n)$, our NTKGP trained baselearners will still have a Bayesian interpretation: $\mathcal{Y}' \leftarrow \Sigma^{-\frac{1}{2}}\mathcal{Y}'$ and $\tilde{f}(\mathcal{X}, \boldsymbol{\theta}) \leftarrow \Sigma^{-\frac{1}{2}}\tilde{f}(\mathcal{X}, \boldsymbol{\theta})$ returns us to the homoscedastic case, where $\Sigma = \text{diag}(\sigma^2(\mathcal{X})) \in \mathbb{R}^{N \times N}$. We will follow Lakshminarayanan et al. [11] and parameterise $\sigma^2(\boldsymbol{x}) = \sigma^2_{\boldsymbol{\theta}}(\boldsymbol{x})$ by an extra output head of the NN, that is trainable alongside the mean function $\mu_{\boldsymbol{\theta}}(\boldsymbol{x})$ when modelling heteroscedasticity.[5]

### 3.4 NTKGP Ensemble Algorithms

We now proceed to train an ensemble of $K$ NTKGP baselearners. Like previous work [11, 24], we independently initialise baselearners, and also use a fixed, independently sampled training set noise $\boldsymbol{\epsilon}_k \in \mathbb{R}^{NC}$ if modelling output noise. These implementation details are all designed to encourage diversity among baselearners, with the goal of approximating the NTKGP posterior predictive for our Bayesian deep ensembles. Appendix F details how to aggregate predictions from trained baselearners. In Algorithm 1, we outline our NTKGP-param method: `data_noise` adds observation noise to targets; `concat` denotes a concatenation operation; and `init`$(\cdot)$ will be standard parameterisation initialisation in the JAX library Neural Tangents [38] unless stated otherwise. As discussed by Pearce et al. [25], there is a choice between "anchoring"/regularising parameters towards their initialisation or an independently sampled parameter set when modelling observation noise. We anchor at initialisation as the linearised NN regime only holds local to parameter initialisation [23], and also this reduces the memory cost of sampling parameters sets. Appendix E details our NTKGP-fn method.

---

**Algorithm 1** NTKGP-param ensemble

---

**Require:** Data $\mathcal{D} = \{\mathcal{X}, \mathcal{Y}\}$, loss function $\mathcal{L}$, NN model $f_{\boldsymbol{\theta}} : \mathcal{X} \to \mathcal{Y}$, Ensemble size $K \in \mathbb{N}$, noise procedure: `data_noise`, NN parameter initialisation scheme: `init`$(\cdot)$
  **for** $k = 1, \dots, K$ **do**
    Form $\{\mathcal{X}_k, \mathcal{Y}_k\} = \text{data\_noise}(\mathcal{D})$
    Initialise $\boldsymbol{\theta}_k \overset{d}{\sim} \text{init}(\cdot)$
    Initialise $\tilde{\boldsymbol{\theta}}_k \overset{d}{\sim} \text{init}(\cdot)$ and denote $\tilde{\boldsymbol{\theta}}_k = \text{concat}(\{\tilde{\boldsymbol{\theta}}_k^{\leq L}, \tilde{\boldsymbol{\theta}}_k^{L+1}\})$
    Set $\boldsymbol{\theta}_k^* = \text{concat}(\{\tilde{\boldsymbol{\theta}}_k^{\leq L}, \mathbf{0}_{p_{L+1}}\})$
    Define $\delta(\boldsymbol{x}) = \nabla_{\boldsymbol{\theta}} f(\boldsymbol{x}, \boldsymbol{\theta}_k)\boldsymbol{\theta}_k^*$
    Define $\tilde{f}_k(\boldsymbol{x}, \boldsymbol{\theta}_t) = f(\boldsymbol{x}, \boldsymbol{\theta}_t) + \delta(\boldsymbol{x})$ and set $\boldsymbol{\theta}_0 = \boldsymbol{\theta}_k$
    Optimise $\mathcal{L}(\tilde{f}_k(\mathcal{X}_k, \boldsymbol{\theta}_t), \mathcal{Y}_k) + \frac{1}{2}\|\boldsymbol{\theta}_t - \boldsymbol{\theta}_k\|_{\Lambda}^2$ for $\boldsymbol{\theta}_t$ to obtain $\hat{\boldsymbol{\theta}}_k$
  **end for**
  **return** ensemble $\{\tilde{f}_k(\cdot, \hat{\boldsymbol{\theta}}_k)\}_{k=1}^K$

---

### 3.5 Classification methodology

For classification, we follow recent works [23, 39, 40] which treat classification as a regression task with one-hot regression targets. In order to obtain probabilistic predictions, we temperature scale our trained ensemble predictions with cross-entropy loss on a held-out validation set, noting that Fong and Holmes [41] established a connection between marginal likelihood maximisation and cross-validation.

Because $\delta(\cdot)$ is untrainable in our NTKGP methods, it is important to match the scale of the NTK $\Theta$ to the scale of the one-hot targets in multi-class classification settings. One can do this either by introducing a scaling factor $\kappa > 0$ such that we scale either: 1) $\tilde{f} \leftarrow \frac{1}{\kappa}\tilde{f}$ so that $\Theta \leftarrow \frac{1}{\kappa^2}\Theta$, or 2) $e_c \leftarrow \kappa e_c$ where $e_c \in \mathbb{R}^C$ is the one-hot vector denoting class $c \leq C$. We choose option 2) for

our implementation, tuning $\kappa$ on a small set of values chosen to match the second moments of the randomly initialised baselearners, in logit space, of each ensemble method on the training set. We found $\kappa$ to be an important hyperparameter that can determine a trade-off between in-distribution and out-of-distribution performance: see Appendix H for further details.

## 4    Experiments

Due to limited space, Appendix I will contain all experimental details not discussed in this section.

**Toy 1D regression task**    We begin with a toy 1D example $y = x\sin(x) + \epsilon$, using homoscedastic $\epsilon \overset{d}{\sim} \mathcal{N}(0, 0.1^2)$. We use a training set of 20 points partitioned into two clusters, in order to detail uncertainty on out-of-distribution test data. For each ensemble method, we use MLP baselearners with two hidden layers of width 512, and erf activation. The choice of erf activation means that both the NTK $\Theta$ and NNGP kernel $\mathcal{K}$ are analytically available [23, 42]. We compare ensemble methods to the analytic GP posterior using either $\Theta$ or $\mathcal{K}$ as prior covariance function using the Neural Tangents library [38].

Figure 1 compares the analytic NTKGP posterior predictive with the analytic NNGP posterior predictive, as well as three different ensemble methods: deep ensembles, RP-param and NTKGP-param. We plot 95% predictive confidence intervals, treating ensembles as one Gaussian predictive distribution with matched moments like Lakshminarayanan et al. [11]. As expected, both NTKGP-param and RP-param ensembles have similar predictive means to the analytic NTKGP posterior. Likewise, we see that only our NTKGP-param ensemble predictive variances match the analytic NTKGP posterior. As foreseen in Proposition 2, the analytic NNGP posterior and other ensemble methods make more confident predictions than the NTKGP posterior, which in this example results in overconfidence on out-of-distribution data.[6]

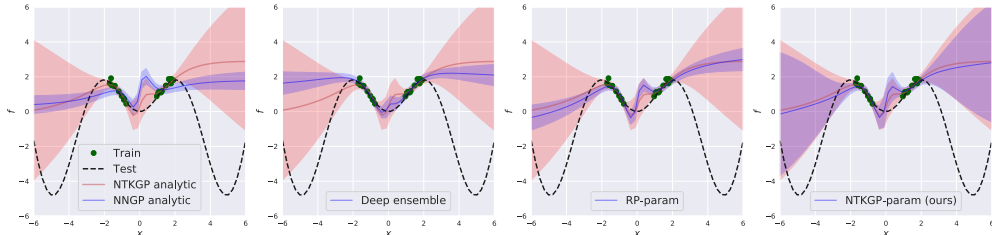

Figure 1: All subplots plot the analytic NTKGP posterior (in red). From left to right, (in blue): analytic NNGP posterior; deep ensembles; RP-param; and NTKGP-param (ours). For each method we plot the mean prediction and 95% predictive confidence interval. Green points denote the training data, and the black dotted line is the true test function $y = x\sin(x)$.

**Flight Delays**    We now compare different ensemble methods on a large scale regression problem using the Flight Delays dataset [43], which is known to contain dataset shift. We train heteroscedastic baselearners on the first 700k data points and test on the next 100k test points at 5 different starting points: 700k, 2m (million), 3m, 4m and 5m. The dataset is ordered chronologically in date through the year 2008, so we expect the NTKGP methods to outperform standard deep ensembles for the later starting points. Figure 2 (Left) confirms our hypothesis. Interestingly, there seems to be a seasonal effect between the 3m and 4m test set that results in stronger performance in the 4m test set than the 3m test set, for ensembles trained on the first 700k data points. We see that our Bayesian deep ensembles perform slightly worse than standard deep ensembles when there is little or no test data shift, but fail more gracefully as the level of dataset shift increases.

Figure 2 (Right) plots confidence versus error for different ensemble methods on the combined test set of 5×100k points. For each precision threshold $\tau$, we plot root-mean-squared error (RMSE) on examples where predictive precision is larger than $\tau$, indicating confidence. As we can see, our NTKGP methods incur lower error over all precision thresholds, and this contrast in performance is magnified for more confident predictions.

**MNIST vs NotMNIST**    We next move onto classification experiments, comparing ensembles trained on MNIST and tested on both MNIST and NotMNIST.[7] Our baselearners are MLPs with

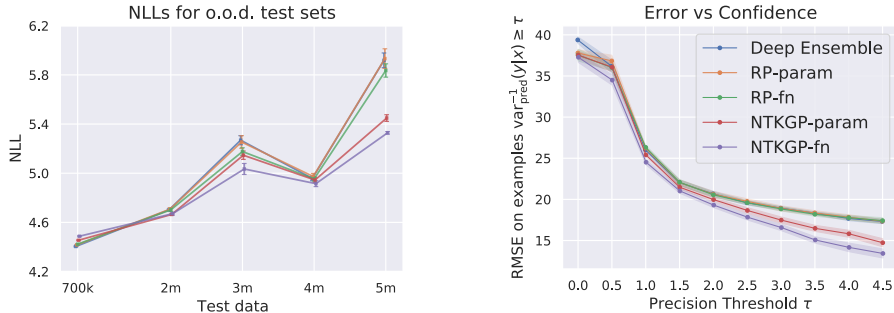

Figure 2: (Left) Flight Delays NLLs for ensemble methods trained on first 700k points of the dataset and tested on various out-of-distribution test sets, with time shift between training set and test set increasing along the $x$-axis. (Right) Error vs Confidence curves for ensembles tested on all $5\times100$k test points combined. Both plots include 95% CIs corresponding to 10 independent ensembles.

2-hidden layers, 200 hidden units per layer and ReLU activation. The weight parameter initialisation variance $\sigma_W^2$ is tuned using the validation accuracy on a small set of values around the He initialisation, $\sigma_W^2$=2, [44] for all classification experiments. Figure 3 shows both in-distribution and out-of-distribution performance across different ensemble methods. In Figure 3 (left), we see that our NTKGP methods suffer from slightly worse in-distribution test performance, with around 0.2% increased error for ensemble size 10. However, in Figure 3 (right), we plot error versus confidence on the combined MNIST and NotMNIST test sets: for each test point $(\boldsymbol{x}, y)$, we calculate the ensemble prediction $p(y = k|\boldsymbol{x})$ and define the predicted label as $\hat{y} = \mathrm{argmax}_k p(y = k|\boldsymbol{x})$, with confidence $p(y = \hat{y}|\boldsymbol{x})$. Like Lakshminarayanan et al. [11], for each confidence threshold $0 \leq \tau \leq 1$, we plot the average error for all test points that are more confident than $\tau$. We count all predictions on the NotMNIST test set to be incorrect. We see in Figure 3 (right) that the NTKGP methods vastly outperform both deep ensembles and RP methods, obtaining over 15% lower error on test points that have confidence $\tau$=0.6, compared to all baselines. This is because our methods correctly make much more conservative predictions on the out-of-distribution NotMNIST test set, as can be seen by Figure 4, which plots histograms of predictive entropies. Due to the simple MLP architecture and ReLU activation, we can compare ensemble methods to analytic NTKGP results in Figures 3 & 4, where we see a close match between the NTKGP ensemble methods (at larger ensemble sizes) and the analytic predictions, both on in-distribution and out-of-distribution performance.

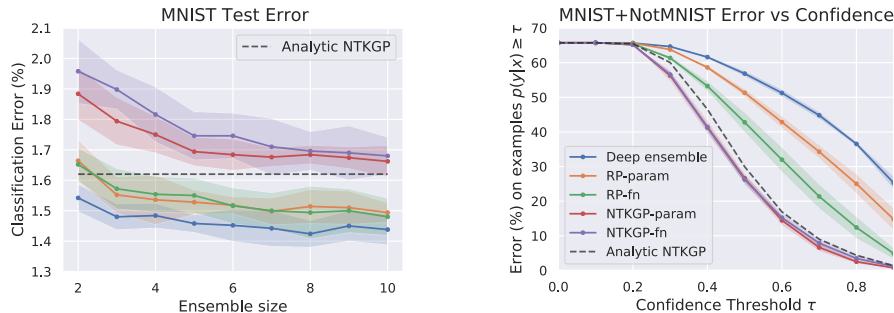

Figure 3: (Left) Classification error on MNIST test set for different ensemble sizes. (Right) Error versus Confidence plots for ensembles, of size 10, trained on MNIST and tested on both MNIST and NotMNIST. CIs correspond to 5 independent runs.

**CIFAR-10 vs SVHN**    Finally, we present results on a larger-scale image classification task: ensembles are trained on CIFAR-10 and tested on both CIFAR-10 and SVHN. We conduct the same setup as for the MNIST vs NotMNIST experiment, with baselearners taking the Myrtle-10 CNN architecture [40] of channel-width 100. Figure 5 compares in distribution and out-of-distribution performance: we see that our NTKGP methods and RP-fn perform best on in-distribution test error. Unlike on the simpler MNIST task, there is no clear difference on the corresponding error versus confidence plot, and this is also reflected in the entropy histograms, which can be found in Figure 8 of Appendix I.

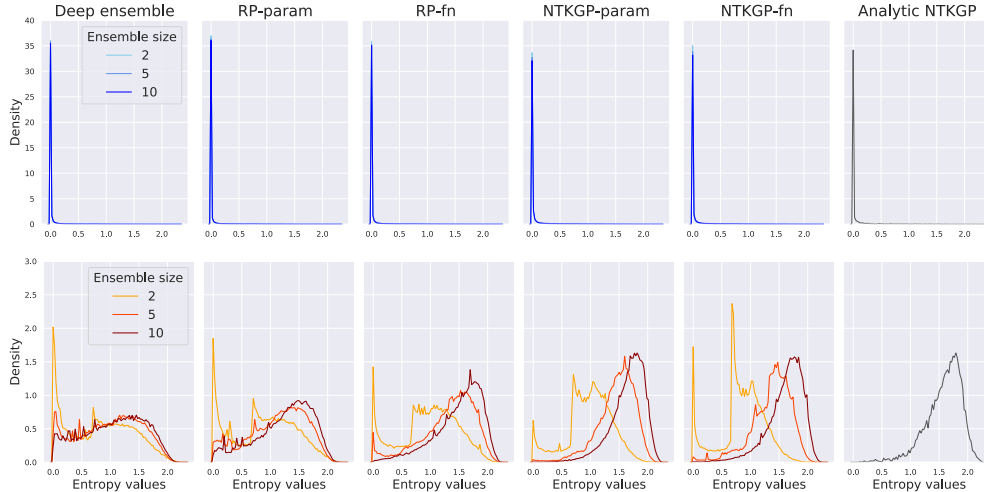

Figure 4: Histograms of predictive entropy on MNIST (top) and NotMNIST (bottom) test sets for different ensemble methods of different ensemble sizes, and also for Analytic NTKGP.

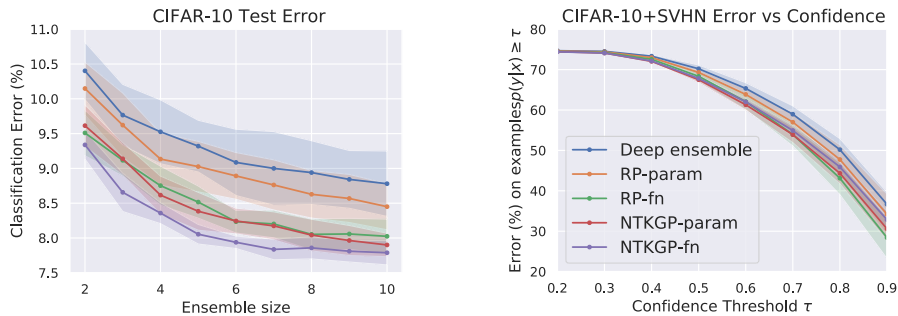

Figure 5: (Left) Classification error on CIFAR-10 test set for different ensemble sizes. (Right) Error versus Confidence plots of ensembles trained on CIFAR-10 and tested on both CIFAR-10 and SVHN. CIs correspond to 5 independent runs.

## 5 Discussion

We built on existing work regarding the Neural Tangent Kernel (NTK), which showed that there is no posterior predictive interpretation to a standard deep ensemble in the infinite width limit. We introduced a simple modification to training that enables a GP posterior predictive interpretation for a wide ensemble, and showed empirically that our Bayesian deep ensembles emulate the analytic posterior predictive when it is available. In addition, we demonstrated that our Bayesian deep ensembles often outperform standard deep ensembles in out-of-distribution settings for both regression and classification tasks.

In terms of limitations, our methods may perform worse than standard deep ensembles [11] when confident predictions are not detrimental, though this can be alleviated via NTK hyperparameter tuning. Moreover, our analyses are planted in the "lazy learning" regime [45, 46], and we have not considered finite-width corrections to the NTK during training [47–49]. In spite of these limitations, the search for a Bayesian interpretation to deep ensembles [11] is of particular relevance to the Bayesian deep learning community, and we believe our contributions provide useful new insights to resolving this problem by examining the limit of infinite-width.

A natural question that emerges from our work is how to tune hyperparameters of the NTK to best capture inductive biases or prior beliefs about the data. Possible lines of enquiry include: the large-depth limit [50], the choice of architecture [51], and the choice of activation [52]. Finally, we would like to assess our Bayesian deep ensembles in non-supervised learning settings, such as active learning or reinforcement learning.

## Broader Impact

We believe that our Bayesian deep ensembles may be useful in situations where predictions that are robust to model misspecification and dataset shift are crucial, such as weather forecasting or medical diagnosis.

## Acknowledgments and Disclosure of Funding

We thank Arnaud Doucet, Edwin Fong, Michael Hutchinson, Lewis Smith, Jasper Snoek, Jascha Sohl-Dickstein and Sheheryar Zaidi, as well as the anonymous reviewers, for helpful discussions and feedback. We also thank the JAX and Neural Tangents teams for their open-source software. BH is supported by the EPSRC and MRC through the OxWaSP CDT programme (EP/L016710/1).

## Footnotes

[1]Throughout this work, the notation "$+h.c.$" means "plus the Hermitian conjugate", like Lee et al. [23].

[3]In this work $\Theta$ always denotes the NTK under NTK parameterisation. It is also possible to model $\Theta$ to be the scaled NTK under standard parameterisation (which depends on layer widths) as in Sohl-Dickstein et al. [32] with minor reweightings to both $\delta(\cdot)$ and, when modelling observation noise, the $L^2$-regularisation described in Appendix D.

[4]With NTK parameterisation, it is easy to see that $\delta(\cdot) \mid \boldsymbol{\theta}_0 \overset{d}{\sim} \mathcal{GP}(0, \hat{\Theta}_0^{\leq L})$, because $\tilde{\boldsymbol{\theta}}^{\leq L} \overset{d}{\sim} \mathcal{N}(0, I_{p - p_{L+1}})$. To extend this to standard parameterisation, note that Eq. (10) is just the first order term in the Taylor expansion of $f(\boldsymbol{x}, \boldsymbol{\theta}_0 + \boldsymbol{\theta}^*)$, which has a parameterisation agnostic distribution, about $\boldsymbol{\theta}_0$.

[5] We use the *sigmoid* function, instead of *softplus* [11], to enforce positivity on $\sigma^2_{\boldsymbol{\theta}}(\cdot)$, because our data will be standardised.

[6]Code for this experiment is available at: `https://github.com/bobby-he/bayesian-ntk`.

[7]Available at `http://yaroslavvb.blogspot.com/2011/09/notmnist-dataset.html`

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
