[Supplementary Material]

# Supplementary Material: Appendix

## Bayesian Deep Ensembles via the Neural Tangent Kernel

## A    Recap of standard and NTK parameterisations

For completeness, we recap the difference between standard and NTK parameterisations & initialisations [22, 23] for an MLP in this section.

Consider an MLP with $L$ hidden layers of widths from $n_0 = d$ to $n_L$ respectively, and final readout layer with $n_{L+1} = C$. For a given $\boldsymbol{x} \in \mathbb{R}^d$, under the *NTK* parameterisation the recurrence relation that constitutes the forward pass of the NN is then:

$$\alpha^{(0)}(\boldsymbol{x}, \boldsymbol{\theta}) = \boldsymbol{x} \tag{13}$$

$$\tilde{\alpha}^{(l+1)}(\boldsymbol{x}, \boldsymbol{\theta}) = \frac{\sigma_W}{\sqrt{n_l}} W^{(l)} \alpha^{(l)}(\boldsymbol{x}, \boldsymbol{\theta}) + \sigma_b b^{(l)} \tag{14}$$

$$\alpha^{(l)}(\boldsymbol{x}, \boldsymbol{\theta}) = \phi(\tilde{\alpha}^{(l)}(\boldsymbol{x}, \boldsymbol{\theta})) \tag{15}$$

for $l \leq L$ where $\tilde{\alpha}^{(l)}$ and $\alpha^{(l)}$ are the preactivations and activations respectively at layer $l$, with entrywise nonlinearity $\phi(\cdot)$. In the NTK parameterion, all parameters $W^{(l)} \in \mathbb{R}^{n_{l+1} \times n_l}$ and $b^{(l)} \in \mathbb{R}^{n_{l+1}}$ for all layers $l$ are initialised as i.i.d. standard normal $\mathcal{N}(0, 1)$. The hyperparameters $\sigma_W$ and $\sigma_b$ are known as the weight and bias variances respectively, and are hyperparameters of the infinite width limit NTK $\Theta$.

On the other hand, under *standard* parameterisation, the recurrence relation of the NN is:

$$\alpha^{(0)}(\boldsymbol{x}, \boldsymbol{\theta}) = \boldsymbol{x} \tag{16}$$

$$\tilde{\alpha}^{(l+1)}(\boldsymbol{x}, \boldsymbol{\theta}) = W^{(l)} \alpha^{(l)}(\boldsymbol{x}, \boldsymbol{\theta}) + b^{(l)} \tag{17}$$

$$\alpha^{(l)}(\boldsymbol{x}, \boldsymbol{\theta}) = \phi(\tilde{\alpha}^{(l)}(\boldsymbol{x}, \boldsymbol{\theta})) \tag{18}$$

with $W_{i,j}^{(l)} \sim \mathcal{N}(0, \frac{1}{n_l}\sigma_W^2)$ and $b_j^{(l)} \sim \mathcal{N}(0, \sigma_b^2)$ at initialisation. Commonly used initialisation schemes like LeCun [53] or He [44] fall into this category.

Regardless of parameterisation, our notation from Sections 2 & 3 corresponds to $f(\boldsymbol{x}, \boldsymbol{\theta}) = \tilde{\alpha}^{(L+1)}(\boldsymbol{x}, \boldsymbol{\theta})$, with $\boldsymbol{\theta} = \{W^{(l)}, b^{(l)}\}_{l=0}^{L}$, $\boldsymbol{\theta}^{\leq L} = \{W^{(l)}, b^{(l)}\}_{l=0}^{L-1}$ and $\boldsymbol{\theta}^{L+1} = \{W^{(L)}, b^{(L)}\}$.

We see that the different parameterisations yield the same distribution for the functional output $f(\cdot, \boldsymbol{\theta})$ at initialisation, but give different scalings to the parameter gradients in the backward pass. Sohl-Dickstein et al. [32] have recently explored further variants of these parameterisations.

## B    Proofs

### B.1    Proof of Proposition 1

**Proposition 1.** $\delta(\cdot) \xrightarrow{d} \mathcal{GP}(0, \Theta^{\leq L})$ *and is independent of* $f_0(\cdot)$ *in the infinite width limit. Thus,* $\tilde{f}_0(\cdot) = f_0(\cdot) + \delta(\cdot) \xrightarrow{d} \mathcal{GP}(0, \Theta)$.

*Proof.* For notational ease, let us define two jointly independent GPs $g(\cdot) \stackrel{d}{\sim} \mathcal{GP}(0, \Theta^{\leq L})$ & $h(\cdot) \stackrel{d}{\sim} \mathcal{GP}(0, \mathcal{K})$. By independence, we have $g(\cdot) + h(\cdot) \stackrel{d}{\sim} \mathcal{GP}(0, \Theta)$. Moreover, let $\delta_m(\cdot), f_m^0(\cdot)$ and $\boldsymbol{\theta}_m^0$ denote $\delta(\cdot), f_0(\cdot)$ and $\boldsymbol{\theta}_0$ respectively at width parameter $m \in \mathbb{N}$. The infinite width limit thus corresponds to $m \to \infty$.

For our purposes, it will be sufficient to prove convergence of finite-dimensional marginals, $(\delta_m(\mathcal{X}), f_m^0(\mathcal{X}')) \xrightarrow{d} (g(\mathcal{X}), h(\mathcal{X}'))$ jointly, for arbitrary sets of inputs $\mathcal{X}, \mathcal{X}'$. Note that previous work [16, 17] has already shown that $f_m^0(\mathcal{X}') \xrightarrow{d} h(\mathcal{X}')$.

The proof that $(\delta_m(\mathcal{X}), f_m^0(\mathcal{X}')) \xrightarrow{d} (g(\mathcal{X}), h(\mathcal{X}'))$ relies on Lévy's Convergence theorem [54] and the Cramér-Wold device (Theorem 29.4 of [55]). Using these results it is sufficient to show, denoting $\varphi_X$ as the characteristic function of a random variable $X$, that:

$$\varphi_{Y_m}(t) \to \varphi_Y(t) \tag{19}$$

where $Y_m = u^\top \delta_m(\mathcal{X}) + v^\top f_m^0(\mathcal{X}')$ and $Y = u^\top g(\mathcal{X}) + v^\top h(\mathcal{X}')$, for all $t \in \mathbb{R}$, $u \in \mathbb{R}^{|\mathcal{X}|C}$ and $v \in \mathbb{R}^{|\mathcal{X}'|C}$. But:

$$\varphi_{Y_m}(t) = \mathbb{E}\big[\exp(itY_m)\big] \tag{20}$$

$$= \mathbb{E}\Big[\mathbb{E}\big[\exp(itY_m) \,\big|\, \boldsymbol{\theta}_{0,m}\big]\Big] \tag{21}$$

$$= \mathbb{E}_{\boldsymbol{\theta}_{0,m}}\big[\exp\big(-t^2 u^\top \hat{\Theta}_{0,m}^{\leq L}(\mathcal{X}, \mathcal{X})u + itv^\top f_m^0(\mathcal{X}')\big)\big] \tag{22}$$

$$= \exp\big(-t^2 u^\top \Theta^{\leq L}(\mathcal{X}, \mathcal{X})u\big)\mathbb{E}_{\boldsymbol{\theta}_{0,m}}\big[\exp\big(itv^\top f_m^0(\mathcal{X}')\big)\big] + r_m \tag{23}$$

$$\to \mathbb{E}\big[\exp(itY)\big] \tag{24}$$

where $r_m$, defined as the difference between Eqs. (23) & (22), can be shown to be $o_m(1)$ using the Bounded Convergence theorem and the empirical NTK convergence results, and by noting that proofs of NTK convergence [22, 23, 28, 29] are all done on a layer-by-layer basis.

The claim that $\tilde{f}_0(\cdot) = f_0(\cdot) + \delta(x) \xrightarrow{d} \mathcal{GP}(0, \Theta)$ then follows by setting $\mathcal{X} = \mathcal{X}'$ and $v = u$.

$\square$

## B.2 Proof of Proposition 2

**Proposition 2.** *For $\sigma^2 = 0$, $\Sigma_{NTKGP} \succeq \Sigma_{DE} \succeq \Sigma_{NNGP}$. Similarly, for $\sigma^2 > 0$, $\Sigma_{NTKGP} \succeq \Sigma_{RP} \succeq \Sigma_{NNGP}$.*

*Proof.* We will prove the case for $\sigma^2 > 0$ as the case for $\sigma^2 = 0$ is similar, and one can replace inversions of $\Theta(\mathcal{X}, \mathcal{X})$ and $\mathcal{K}(\mathcal{X}, \mathcal{X})$ with generalised inverses if need be.

Let $\mathcal{X}'$ be an arbitrary test set. We will first show $\Sigma_{RP} \succeq \Sigma_{NNGP}$. It will suffice to show that $\Sigma_{RP}(\mathcal{X}', \mathcal{X}') - \Sigma_{NNGP}(\mathcal{X}', \mathcal{X}') \succeq 0$ is a p.s.d. matrix. But it is not hard to check that:

$$\Sigma_{RP}(\mathcal{X}', \mathcal{X}') - \Sigma_{NNGP}(\mathcal{X}', \mathcal{X}') = U(\mathcal{K}(\mathcal{X}, \mathcal{X}) + \sigma^2 I)U^\top \tag{25}$$

which is clearly p.s.d, where $U = \Theta(\mathcal{X}', \mathcal{X})(\Theta(\mathcal{X}, \mathcal{X}) + \sigma^2 I)^{-1} - \mathcal{K}(\mathcal{X}', \mathcal{X})(\mathcal{K}(\mathcal{X}, \mathcal{X}) + \sigma^2 I)^{-1} \in \mathbb{R}^{|\mathcal{X}'| \times |\mathcal{X}|}$

Likewise, to show $\Sigma_{NTK} \succeq \Sigma_{RP}$ we can check that:

$$\Sigma_{NTK}(\mathcal{X}', \mathcal{X}') - \Sigma_{RP}(\mathcal{X}', \mathcal{X}') = U_1 + U_2 \Delta(\mathcal{X}, \mathcal{X})U_2^\top \succeq 0 \tag{26}$$

where

$$U_1 = \Delta(\mathcal{X}', \mathcal{X}') - \Delta(\mathcal{X}', \mathcal{X})\Delta^g(\mathcal{X}, \mathcal{X})\Delta(\mathcal{X}, \mathcal{X}') \tag{27}$$

and $\Delta = \Theta^{\leq L} \succeq 0$ is the contributions to the NTK from parameters before the final layer as before. Finally, we need to define $U_2$ as:

$$U_2 = \Theta(\mathcal{X}', \mathcal{X})(\Theta(\mathcal{X}, \mathcal{X}) + \sigma^2 I)^{-1} - \Delta(\mathcal{X}', \mathcal{X})\Delta^g(\mathcal{X}, \mathcal{X}) \tag{28}$$

The notation $\Delta^g(\mathcal{X}, \mathcal{X})$ denotes the generalised inverse. $U_1 \succeq 0$ follows from standard properties of generalised Schur complements, as does the fact that $\Delta(\mathcal{X}', \mathcal{X})\Delta^g(\mathcal{X}, \mathcal{X})\Delta(\mathcal{X}, \mathcal{X}) = \Delta(\mathcal{X}', \mathcal{X})$, which is required for Eq. (26) to hold.

$\square$

# C  Alternative constructions of NTKGP baselearners

To summarise the analysis in Section 3, the criteria for an NTKGP baselearner $\tilde{f}(\cdot, \boldsymbol{\theta})$ is that:
1) $\tilde{f}(\cdot, \boldsymbol{\theta}_0) \xrightarrow{d} \mathcal{GP}(0, \Theta)$ as width increases, while 2) preserving the initial Jacobian $\nabla_{\boldsymbol{\theta}} f_0(\cdot) = \nabla_{\boldsymbol{\theta}} \tilde{f}_0(\cdot)$.

A possible alternative construction would be if one could (approximately) sample a fixed $f^* \overset{d}{\sim} \mathcal{GP}(0, \Theta)$, and set:

$$\tilde{f}_t(\cdot) = f_t(\cdot) + f^*(\cdot) - f_0(\cdot) \tag{29}$$

It is easy to approximately sample $f^*$ for finite width NNs using a single JVP, under either standard or NTK parameterisation, by sampling $\tilde{\theta}$ independent of $\theta_0$ and setting:

$$f^*(\boldsymbol{x}) = \nabla_{\boldsymbol{\theta}} f(\boldsymbol{x}, \boldsymbol{\theta}_0) \tilde{\boldsymbol{\theta}} \tag{30}$$

Note that Eq. (29) requires computation of two forward passes $f_t$ and $f_0$ in addition to a JVP $\nabla_{\boldsymbol{\theta}} f(\boldsymbol{x}, \boldsymbol{\theta}_0) \tilde{\boldsymbol{\theta}}$. For some implementations of JVPs, such as in JAX [35], the computation of $f_0$ will come essentially for free alongside the computation of $\nabla_{\boldsymbol{\theta}} f(\boldsymbol{x}, \boldsymbol{\theta}_0) \tilde{\boldsymbol{\theta}}$, because the JVP is centered about the same "primal" parameters $\boldsymbol{\theta}_0$ that are used for $f_0$. Hence, this alternative $\tilde{f}$ presented in Eq. (29) may have similar costs to our main construction in Section 3, for certain AD packages.

A second valid alternative to $\tilde{f}_t$ would be to replace $f_t$ with $f_t^{\text{lin}}$, which would give $\tilde{f}^{\text{lin}}(\boldsymbol{x}, \boldsymbol{\theta}_t) = \nabla_{\boldsymbol{\theta}} f(\boldsymbol{x}, \tilde{\boldsymbol{\theta}}) \boldsymbol{\theta}_t$ (where we swap $\tilde{\boldsymbol{\theta}}$ and $\boldsymbol{\theta}_0$ for notational consistency with other NTKGP methods, and initialise at $\boldsymbol{\theta}_0$). Because $\tilde{\boldsymbol{\theta}}$ is fixed, we see that $\tilde{f}^{\text{lin}}(\cdot, \boldsymbol{\theta}_t)$ is linear in $\boldsymbol{\theta}_t$. This gives a realisation of the "sample-then-optimize" approach [34] to give posterior samples from randomly initialised linear models, and ensures that $\tilde{f}_\infty^{\text{lin}}(\cdot)$ is an exact posterior sample (using the empirical NTK $\hat{\Theta}_0$ as prior kernel) irrespective of parameterisation or width. Note though, of course, the linearised regime holds for $\tilde{f}_t^{\text{lin}}$ throughout parameter space, hence for strongly convex optimisation problems like regression tasks with observation noise, the initialisation is irrelevant. We will call $\tilde{f}_\infty^{\text{lin}}$ trained in such a way an *NTKGP-Lin* baselearner.

## D    Regularisation in the NTKGP and RP training procedures

As stated in Lemma 3 of Osband et al. [24], suppose we are in a Bayesian linear regression setting with linear map $g_{\boldsymbol{\theta}}(\boldsymbol{z}) = \boldsymbol{z}^\top \boldsymbol{\theta}$, model $y = g_{\boldsymbol{\theta}}(\boldsymbol{z}) + \epsilon$ for $\epsilon \sim \mathcal{N}(0, \sigma^2)$ i.i.d., and parameter prior $\boldsymbol{\theta} \sim N(0, \lambda I_p)$. Then, having observed training data $\{(\boldsymbol{z}_i, y_i)\}_{i=1}^n$, solving the following optimisation problem returns a posterior sample $\boldsymbol{\theta}$:

$$\tilde{\boldsymbol{\theta}} + \operatorname*{argmin}_{\boldsymbol{\theta}} \sum_{i=1}^n \frac{1}{2\sigma^2} \left\| \tilde{y}_i - (g_{\boldsymbol{\theta}} + g_{\tilde{\boldsymbol{\theta}}})(\boldsymbol{z}_i) \right\|_2^2 + \frac{1}{2\lambda} \|\boldsymbol{\theta}\|_2^2 \tag{31}$$

where $\tilde{y}_i \sim \mathcal{N}(y_i, \sigma^2)$ and $\tilde{\boldsymbol{\theta}} \sim \mathcal{N}(0, \lambda I_p)$.

We see that when there is a homoscedastic prior $\mathcal{N}(0, \lambda I_p)$ for $\boldsymbol{\theta}$ that the correct weighting of $L^2$ regularisation is $\|\boldsymbol{\theta}\|_\Lambda^2 = \frac{1}{\lambda} \boldsymbol{\theta}^\top \boldsymbol{\theta}$. In fact, even with a heteroscedastic prior $\boldsymbol{\theta} \sim \mathcal{N}(0, \Lambda)$ with a diagonal matrix $\Lambda \in \mathbb{R}_+^{p \times p}$ and diagonal entries $\{\lambda_j\}_{j=1}^p$, it is straightforward to show that the correct setting of regularisation is $\|\boldsymbol{\theta}\|_\Lambda^2 = \boldsymbol{\theta}^\top \Lambda^{-1} \boldsymbol{\theta}$ in order to obtain a posterior sample of $\boldsymbol{\theta}$. For RP-param or NTKGP-param methods, with initial parameters $\boldsymbol{\theta}_0$, we have regularisation $\|\boldsymbol{\theta} - \boldsymbol{\theta}_0\|_\Lambda^2 = (\boldsymbol{\theta} - \boldsymbol{\theta}_0)^\top \Lambda^{-1} (\boldsymbol{\theta} - \boldsymbol{\theta}_0)$, which can be seen as a Mahalanobis distance.

For an NN in the linearised regime [23], this is related to the fact that the NTK and standard parameterisations initialise parameters differently, yet yield the same functional distribution for a randomly initialised NN. In the standard parameterisation, $\lambda_j$ will be a factor of the NN width smaller than in the NTK parameterisation, but the corresponding feature map $\boldsymbol{z}$ will be a square root factor of the NN width larger. Thus, solving Eq. (31) will lead to the same functional outputs in both parameterisations, if the NN remains in the linearised regime. However, only with our NTKGP trained baselearners $\tilde{f}$ do you get a posterior interpretation to the trained NN because of the difference between the NNGP and the NTK that standard training does not account for, and because the linearised regime only holds locally to the parameter initialisation.

## E    Additional ensemble algorithms

Here, we present our ensemble algorithms for NTKGP-Lin (Algorithm 2) and NTKGP-fn (Algorithm 3), to complement the NTKGP-param algorithm that was presented in Section 3.4.

---

**Algorithm 2** NTKGP-Lin ensemble

---

**Require:** Data $\mathcal{D} = \{\mathcal{X}, \mathcal{Y}\}$, loss function $\mathcal{L}$, NN model $f_{\boldsymbol{\theta}} : \mathcal{X} \to \mathcal{Y}$, Ensemble size $K \in \mathbb{N}$, noise
  procedure: `data_noise`, NN parameter initialisation scheme: `init(·)`
  **for** $k = 1, \ldots, K$ **do**
    Form $\{\mathcal{X}_k, \mathcal{Y}_k\} = $ `data_noise`$(\mathcal{D})$
    Initialise $\boldsymbol{\theta}_k \overset{d}{\sim}$ `init(·)`
    Initialise $\tilde{\boldsymbol{\theta}}_k \overset{d}{\sim}$ `init(·)`
    Define $\tilde{f}_k^{\text{lin}}(\boldsymbol{x}, \boldsymbol{\theta}_t) = \nabla_{\boldsymbol{\theta}} f(\boldsymbol{x}, \tilde{\boldsymbol{\theta}}_k)\boldsymbol{\theta}_t$ and set $\boldsymbol{\theta}_0 = \boldsymbol{\theta}_k$
    Optimise $\mathcal{L}(\tilde{f}_k^{\text{lin}}(\mathcal{X}_k, \boldsymbol{\theta}_t), \mathcal{Y}_k) + \frac{1}{2}\|\boldsymbol{\theta}_t - \boldsymbol{\theta}_k\|_{\Lambda}^2$ for $\boldsymbol{\theta}_t$ to obtain $\hat{\boldsymbol{\theta}}_k$
  **end for**
  **return** ensemble $\{\tilde{f}_k^{\text{lin}}(\cdot, \hat{\boldsymbol{\theta}}_k)\}_{k=1}^K$

---

---

**Algorithm 3** NTKGP-fn ensemble

---

**Require:** Data $\mathcal{D} = \{\mathcal{X}, \mathcal{Y}\}$, loss function $\mathcal{L}$, NN model $f_{\boldsymbol{\theta}} : \mathcal{X} \to \mathcal{Y}$, Ensemble size $K \in \mathbb{N}$, noise
  procedure: `data_noise`, NN parameter initialisation scheme: `init(·)`
  **for** $k = 1, \ldots, K$ **do**
    Form $\{\mathcal{X}_k, \mathcal{Y}_k\} = $ `data_noise`$(\mathcal{D})$
    Initialise $\boldsymbol{\theta}_k \overset{d}{\sim}$ `init(·)`
    Initialise $\tilde{\boldsymbol{\theta}}_k \overset{d}{\sim}$ `init(·)` and denote $\tilde{\boldsymbol{\theta}}_k = $ `concat`$(\{\tilde{\boldsymbol{\theta}}_k^{\leq L}, \tilde{\boldsymbol{\theta}}_k^{L+1}\})$
    Set $\boldsymbol{\theta}_k^* = $ `concat`$(\{\sqrt{2}\tilde{\boldsymbol{\theta}}_k^{\leq L}, \tilde{\boldsymbol{\theta}}_k^{L+1}\})$
    Define $\delta(\boldsymbol{x}) = \nabla_{\boldsymbol{\theta}} f(\boldsymbol{x}, \tilde{\boldsymbol{\theta}}_k)\boldsymbol{\theta}_k^*$
    Define $\tilde{f}_k(\boldsymbol{x}, \boldsymbol{\theta}_t) = f(\boldsymbol{x}, \boldsymbol{\theta}_t) + \delta(\boldsymbol{x})$ and set $\boldsymbol{\theta}_0 = \boldsymbol{\theta}_k$
    Optimise $\mathcal{L}(\tilde{f}_k(\mathcal{X}_k, \boldsymbol{\theta}_t), \mathcal{Y}_k) + \frac{1}{2}\|\boldsymbol{\theta}_t\|_{\Lambda}^2$ for $\boldsymbol{\theta}_t$ to obtain $\hat{\boldsymbol{\theta}}_k$
  **end for**
  **return** ensemble $\{\tilde{f}_k(\cdot, \hat{\boldsymbol{\theta}}_k)\}_{k=1}^K$

---

In Algorithm 3 we seek to reinitialise $\tilde{f}_k(\boldsymbol{x}, \boldsymbol{\theta}_0)$ from $\mathcal{GP}(0, \mathcal{K})$ to $\mathcal{GP}(0, 2\Theta)$ in the infinite width limit, following the randomised prior function method of Osband et al. [24]. While there are many ways to do this we choose to use only one JVP, with a reweighted tangent vector, for $\delta(\cdot)$ in order to reduce extra computational costs. It would be similarly possible to model a scaling factor $\beta$ for the prior function, like [24], using a single JVP with a differently reweighted tangent vector.

Note also that for the NTKGP-fn it is unreasonable to assume that the linearised NN dynamics will hold true for the duration of training because, unlike in NTKGP-param (Algorithm 1) we regularise towards the origin not the initialised parameters.

## F  Aggregating predictions from ensemble members

For completeness, we now describe how to aggregate predictions from ensemble members. Given a test point $(\boldsymbol{x}, y)$, for each baselearner NN $k \leq K$, we suppose we have a probabilistic prediction $p_k(y|\boldsymbol{x})$ obtained from the NN output. We then treat the ensemble as a uniformly-weighted mixture model over baselearners and combine predictions as $p(y|\boldsymbol{x}) = \frac{1}{K}\sum_{k=1}^K p_k(y|\boldsymbol{x})$. For our Bayesian deep ensembles, we can view this aggregation as a Monte Carlo approximation of the GP posterior predictive with NTK prior.

For classification tasks, this aggregation is exactly an average of predicted probabilities. For regression tasks, the prediction is a mixture of normal distributions, and we follow Lakshminarayanan et al. [11] by approximating the ensembled prediction as a single Gaussian with matched moments. That is to say, if $p_k(y|\boldsymbol{x}) \sim \mathcal{N}(\mu_k(\boldsymbol{x}), \sigma_k^2(\boldsymbol{x}))$, then we approximate $p(y|\boldsymbol{x})$ by $\mathcal{N}(\mu_*(\boldsymbol{x}), \sigma_*^2(\boldsymbol{x}))$ for $\mu_*(\boldsymbol{x}) = \frac{1}{K}\sum_k \mu_k(\boldsymbol{x})$ and $\sigma_*^2(\boldsymbol{x}) = \frac{1}{K}\sum_k (\mu_k^2(\boldsymbol{x}) - \mu_*^2(\boldsymbol{x})) + \sigma_k^2(\boldsymbol{x})$.

## G  Comparison of memory and computation costs for ensemble methods

There is only a negligible training-time computational overhead for our NTKGP methods compared to other ensemble methods [11, 24], for a training set of fixed size (e.g. MNIST, CIFAR-10). This

is because one can obtain and store our fixed additive JVPs $\delta$ in a single pass over the training data. For test-time constrained applications, one can employ ensemble distillation [56] for our NTKGP ensembles as one would for standard deep ensembles.

For completeness, we include in Table 2 (left) the computational cost of different ensemble methods when the modified forward pass $\tilde{f}$ needs to be computed on the fly for new data, though we again stress that this is not necessary for train nor test time, as described in the paragraph above. A rule of thumb for a library offering forward-mode AD, like JAX [35], is that a JVP costs on the order of three standard forward passes in terms of FLOPs. We use forward-mode AD to compute JVPs as this is known to be more memory-efficient than reverse-mode AD for JVP computation. It is worth pointing out that our methods share the same trainable parameters as standard deep ensembles, and so do not incur any additional computational cost in the backward pass.

Table 2: Comparison of computational and memory costs of different ensemble methods per ensemble member. Computational costs are specified per (modified) forward pass and represent a naive worst-case scenario (presented for completeness); a more astute approach renders only a negligible difference between ensemble methods, as discussed in this section.

| Method | Computational cost | | Parameter sets to store | |
|---|---|---|---|---|
| | Forward passes | JVPs | Train time | Test time |
| Deep ensembles | 1 | 0 | 1 | 1 |
| RP-param | 1 | 0 | 2 | 1 |
| RP-fn | 2 | 0 | 2 | 2 |
| NTKGP-param | 1 | 1 | 3 | 3 |
| NTKGP-fn | 1 | 1 | 3 | 3 |

In terms of memory, both NTKGP and RP methods require storage of extra sets of parameters in order to compute the untrainable additive functions $\delta(\cdot)$ and regularise in parameter space, displayed in Table 2 (right). However, the activations of the extra forward pass in the Randomised prior function method need not be stored. And moreover, forward mode JVPs are composed alongside the primitive operations that comprise the forward pass, so the memory requirements incurred by the extra JVP are independent of the NN depth for our NTKGP methods. Note that the memory bottleneck for large NNs is most often from the need to store activations for the backward pass [57] and not from storing parameter sets, hence our NTKGP ensembles are not affected by the main memory bottleneck for large NNs, relative to standard deep ensembles.

It is worth noting that our Bayesian deep ensembles still retain the distributability of standard deep ensembles. Moreover, our computational and memory costs still scale linearly in dataset size and parameter space dimension, enabling us to work with large scale datasets like Flight Delays [43].

Finally, in this section we only compare the costs associated to different ensemble methods. Ensembles methods are known to be computationally expensive and there has been recent interest in the community to derive new methods [58, 59] that reduce such costs. However, at the time of writing, deep ensembles [11] are state-of-the-art for uncertainty quantification tasks [12], and hence we believe a comparison of costs between ensemble methods is most appropriate for this work.

## H  Scaling for one-hot targets in classification

As discussed in Section 3.5 and repeated here for completeness: because $\delta(\cdot)$ is untrainable in our NTKGP methods, it is important to match the scale of the NTK $\Theta$ to the scale of the one-hot targets in multi-class classification settings. One can do this either by introducing a scaling factor $\kappa > 0$ such that we scale either: 1) $\tilde{f} \leftarrow \frac{1}{\kappa}\tilde{f}$ and so $\Theta \leftarrow \frac{1}{\kappa^2}\Theta$, or 2) $e_c \leftarrow \kappa e_c$ where $e_c \in \mathbb{R}^C$ is the one-hot vector denoting class $c \leq C$. We choose option 2) for our implementation.

To set $\kappa$, for each ensemble method we calculated the mean squared values of baselearner outputs at initialisation, which we define for convenience as $\zeta_0$, on the training set for that particular ensemble method, and tuned $\kappa^2$ (based on validation accuracy) on a small linear scale centered around $C\zeta_0$, where $C$ is the number of classes. This is in order to match the second moments of the random NNs at initialisation with the scaled one-hot targets across the $C$ classes. For example, for NTKGP-param, we set $\zeta_0 = \frac{1}{|\mathcal{X}|}\sum_{\boldsymbol{x}\in\mathcal{X}}\Theta(\boldsymbol{x},\boldsymbol{x}) \in \mathbb{R}^+$.

To illustrate the importance of $\kappa$, in Figure 6 we present the corresponding results to Figure 5 where instead of setting $\kappa$ dependent on the scale of each ensemble methods' initialised baselearners, as above, we set $\kappa = \frac{1}{|\mathcal{X}|}\sum_{\boldsymbol{x}\in\mathcal{X}}\Theta(\boldsymbol{x},\boldsymbol{x})\in\mathbb{R}$ for all ensemble methods. This is the base $\kappa$ value for NTKGP-param at initialisation, but note that we did not tune neither $\kappa$ (around this base value) nor weight variance (set at $\sigma_W^2 = 2$ like He initialisation [44], which has been optimised for standard NNs and hence standard deep ensembles) for Figure 6.

Figure 6: Figure 5 but where regression target scale $\kappa$ is constant across ensemble methods and set to match the second moment of the NTK on the training set at initialisation. Error bars correspond to 5 independent runs.

In Figure 6 we see a different results to Figure 5, as here our NTKGP methods suffer slightly on in-distribution performance but also outperform the baselines methods on out-of-distribution detection. This highlights the importance of the regression target scale when considering classification tasks, and moreover reflects a general theme in our experiments of the trade-off between more aggressive predictions (that tend to perform better on in-distribution) and more conservative predictions (that tend to perform better on out-of-distribution). In our classification methodology, larger $\kappa$ values lead to more confident predictions. We point out that this is an issue that affects all ensemble methods for uncertainty quantification and is not limited to our Bayesian ensembles.

# I  Experimental Details & additional plots

## I.1  Toy 1d example

We set ensemble size $K = 20$, and train on full batch GD with learning rate 0.001 for 50,000 iterations under standard parameterisation in Neural Tangents [38], with $\sigma_W = 1.5$ & $\sigma_b = 0.05$, for $\sigma_W, \sigma_b$ defined as in Appendix A. In Figure 7 we evaluate the impact of the ensemble size on this toy problem for different ensemble methods. We find that, of the two methods that approximate the analytic NTKGP mean predictor (c.f. Table 1), the analytic mean approximation of NTKGP-param degrades compared to RP-param at small ensemble sizes, although the predictive uncertainties are well matched even at small ensemble sizes. The degradation in mean predictor is unsurprising as there is more (untrainable) noise in the initialised NTKGP baselearners. One simple possible solution to this problem, which we leave for future work, is to use separate baselearners for the mean and uncertainty predictions, like in Ciosek et al. [26].

## I.2  Flight Delays

Our baselearners are MLPs with 4 hidden layers, 100 hidden units per layer and ReLU activations, and we use standard parameterisation with $\sigma_W = 1$ & $\sigma_b = 0.05$, and choose ensemble size $K = 5$. We train for 10 epochs with learning rate 0.001, batch size 100 and Adam [60]. For all experiments, all ensemble methods apart from standard deep ensembles [11] are $L^2$ regularised according to Appendix D, with weight decay strength set to $10^{-4}$ for standard deep ensembles.

We use a validation set of size 50k that is sampled uniformly from the training set of size 700k, and early stop baselearner NNs based on validation set loss. Inputs and targets are standardised so that the training data is zero mean and unit variance.

## I.3  MNIST vs. NotMNIST

For all image classification experiments, we use a $90-10\%$ split for the train-validation sets needed for temperature scaling.

Figure 7: Comparison between ensemble methods (blue) and the analytic NTKGP posterior as the ensemble size is varied on toy example.

Baselearners are MLPs with 2-hidden layers, 200 hidden units per layer and ReLU activations. We standardise data to have mean 0 and standard deviation 1 across flattened pixels.

For all ensemble methods, we use standard parameterisation with fixed bias standard deviation $\sigma_b = 0.05$, observation noise $\sigma = 0.1$ and tune weight variance $\sigma_W^2$ on a small linear scale around $\sigma_W^2 = 2$. We set observation noise $\sigma = 0.1$ for . We train for 20 epochs with batch size 100, learning rate 0.001 and Adam [60]. We do not early stop for any classification experiment, and use the final trained baselearners throughout.

For the analytic NTKGP results, we use the NTK in NTK parameterisation, and use the same observation noise and bias variance as for ensemble methods. However, we fix $\sigma_W^2 = 2$ and also do not tune target scale $\kappa$ (set to the base value described in Appendix A) due to computational resources. We also use only half the test sets both for MNIST and NotMNIST due to resource requirements, keeping the ratios of test sizes consistent in order for the error versus confidence plot Figure 3 (right) to be comparable. To compute test and out-of-distribution predictions, having obtained the optimal temperature scale $T^*$ and analytic NTKGP predictions in logit space, $p(\cdot|\mathcal{X}, \mathcal{Y})$, we approximate the softmax class probability predictions: $\int \text{softmax}(z/T^*)p(dz|\mathcal{X}, \mathcal{Y})$, by a Monte Carlo ensemble approximation with 100 samples.

For all classification ensemble methods, we temperature scale on validation cross entropy for 5 epochs with batch size 100 and learning rate 0.1, whereas for analytic NTKGP we temperature scale for 1000 epochs on full batch size 6000. Like above, we approximate the analytic NTKGP validation predictions (for temperature scaling) by a Monte Carlo ensemble, this time of size 10. We found the various temperature scaling training hyperparameter considerations here to be unimportant to achieve convergence, due to the fact that the temperature scale is a scalar value.

## I.4 CIFAR-10 vs SVHN

Baselearners are Myrtle-10 CNNs [40] with 100 channel width and ReLU activations. We use $\sigma_b = 0.01$ and set observation noise $\sigma = 0.1$. Like for MNIST we tune $\sigma_W^2$ on a small linear scale around $\sigma_W^2 = 2$. We train using SGD, with momentum parameter 0.9, for 100 epochs and learning rate 0.001, which is decayed to 0.0002 after 80 epochs. In the first 5 epochs we raise the learning rate in linear increments from 0.0001 to 0.001. We use batch size 125. During training we apply random crops and horizontal flips before standardisation. We do not compare to the analytic NTKGP for the Myrtle-10 CNN due to resource requirements.

Figure 8 displays entropy histograms for ensembles trained on CIFAR-10 and tested on in distribution CIFAR-10 test data and out-of-distribution SVHN test data, corresponding to the same experiments as in Figure 5. As we can see, there is a much less noticeable difference between ensemble methods compared to the simpler MNIST vs NotMNIST case.

Figure 8: Histograms of predictive entropy on CIFAR-10 (top) and SVHN (bottom) test sets for different ensemble methods and for different ensemble sizes.