[Reviews · NeurIPS 2020]

Review 1

Summary and Contributions: This paper introduces an extra randomly-initialized-then-fixed function that is added to the neural networks for deep ensembles training. The paper shows that the proposed scheme yields some posterior predictive distribution (NTKGP) in the infinite width limit. The distribution is shown to have larger variance (more conservative predictions) than standard deep ensembles.

Strengths: Significance: The paper proposes a straightforward random function to be added to the neural network so that when the width goes to infinity, the distribution of randomly initialized NN converges to a Gaussian prior whose covariance function coincides with the NTK. Then through initializing multiple networks and training them individually, one can obtain an approximation to the Bayesian posterior of that NTK prior.

Weaknesses: Theoretical grounding: The two propositions of the paper seems straightforward. Most of the theoretical work is from Lee at al.'s paper. Empirical evaluation: I am not sure where do the benefits come from when the distribution of the ensemble members has a GP posterior interpretation. Empirically speaking, the proposed method may not worth a try considering its marginal improvement and significant computational overhead. Significance: The major weakness is the significance of the paper. I am not sure where do the benefits come from when the distribution of the ensemble members has a GP posterior interpretation. It is better that the authors address the issue in the paper.

Correctness: I did not carefully check the correctness of the propositions and the experiments.

Clarity: Overall the paper is well written. The major problem is the significance of the paper. I am not sure where do the benefits come from when the distribution of the ensemble members has a GP posterior interpretation. It is better that the authors address the issue in the paper. Besides I have the following concerns. - Line 128: "the NTK parameterisation" and "standard parameterisations". The two terms are not introduced in the paper.

Relation to Prior Work: Yes. The Table 1 summarizes the distributions of four methods.

Reproducibility: Yes

Additional Feedback: === update === I have read the other reviews and the author response. I prefer not to change my opinion.


Review 2

Summary and Contributions: The paper is concerned with the interpretation of deep ensembles as approximations to posterior predictive distributions in Bayesian neural networks. In particular, the jumping off point for this work is a formal argument, based on recent work on neural tangent kernel (NTK), that standard deep ensembles do not have valid interpretations as Bayesian posterior predictive approximations. This point is particularly relevant to the Bayesian deep learning (BDL) literature, as several papers have recently demonstrated deep ensembles' tendency to outperform more principled approximations to Bayesian posterior predictions (e.g., VI and even MCMC methods) on tasks requiring calibrated uncertainty. As such, understanding the relationship between deep ensembles and posterior inference is of particular relevance. The authors proceed to analyse the predictive distribution of the NTK, and propose to simply add the missing terms to the function output such that the completion provides a parameterisation with a valid interpretation as a posterior for the desired kernel. The correction term is such that trained neural networks can be interpreted as exact samples from the desired posterior distribution, which in turn allows us to interpret predictions from the resulting ensembles as approximations to the Bayesian posterior predictive distribution. The authors then proceed to flesh out this procedure for different noise models, including the heteroskedastic noise case, which is most appropriate for regression tasks. Finally, the authors demonstrate the utility of their proposed correction on several experiments, including toy data, a benchmark regression dataset, and classification of MNIST (and NotMNIST) digits. Post discussion update: --------------------------- I have reviewed the authors' responses and the additional reviews. Following this, and the reviewer discussions, I am still quite strongly in favour of accepting the paper based on its contributions. I disagree with some of the stated concerns regarding the technical / conceptual contributions of the paper, as well as with concerns regarding it practical applicability. 1. Conceptual contributions: several papers have recently shown that, on a suite of tasks regarding calibrated uncertainty, ensemble based methods tend to consistently outperform more principled approximations to posterior inference. It has also been claimed that ensemble methods can be viewed as principled approximations. The central contribution of this paper, as I see it, is to address and make steps towards resolving these questions. Namely, using the NTK limiting behaviour we can see precisely that (1) standard ensembles cannot be interpreted as principled approximations to the posterior, and (2) there is a "simple" modification to the training procedure such that ensemble members _can_ be interpreted as exact samples from the posterior (in the infinite width limit). The resolution of this question is an important contribution. The provided propositions, whether straightforward or not, are useful in validating the claims made by the authors. 2. Practical applications: the authors provide several experiments that demonstrate the usefulness of the proposed modification in relevant tasks and settings. More so when considering the additional experiments provided in the rebuttal. I am less worried about standard "large-scale" classification experiments. The authors (correctly, in my opinion) focus instead on several experiments where calibrated uncertainty is of particular importance (such as the o.o.d. tasks), which are better suited to evaluate the properties of the proposed method. I agree that scalability is a question that should be addressed, but on further reading of the paper and the authors' rebuttal, I am less concerned about this as the computation overhead is "negligible compared to standard deep ensembles...". As such, I am maintaining my score, and endorse the acceptance of the paper.

Strengths: Overall, I think this is a very strong paper. The paper is very well motivated, and deals with a point that is of particular relevance to the recent BDL literature. The technical derivations appear sound, and though I am not an expert regarding the NTK (and therefore have trouble assessing the correctness of the derivations), the arguments are very clear and logical. One of the key strengths of this work in my opinion is that it provides a formal treatment of the relationship between deep ensembles and Bayesian posterior predictive distributions. This is particularly relevant as there have been several questions around this point in the literature recently, both empirically e.g. ([12, a]) and conceptually [14]. Further, the authors tackle many of the important questions regarding parameterisations and handling different observation noise models, which is often overlooked in the BDL literature. Finally, the authors provide empirical evidence that their proposed correction leads to significant improvements in the posterior predictive distribution over sensible baseline (both standard deep ensembles and more sophisticated approximations using randomised priors [22]) in both regression and classification settings. [a] Wenzel et al. How good is the Bayes posterior in deep neural networks really? 2020.

Weaknesses: 1. As far as I am able to discern, the main weakness of the proposed method is its computational overhead. In particular, the correction term requires evaluating a Jacobian-vector product, which is required during both training and test time. This is a significant level of added complexity, both in terms of the forward pass itself and the computational burden of training / evaluation. As such, it is not clear to me exactly how scalable this method will be, and whether it will be feasible to compare it to simpler ensemble methods, even at moderate-scale tasks such as CIFAR10/100 classification. 2. In the flight delays experiment, we see that deep ensembles (and in fact, all other baseline methods) outperform the NTKGP when there is no dataset shift. This appears surprising, especially in light of the results shown in Fig 1. This would be easier to understand if the $y$-axis in Fig 2 (left) was MSE, but for NLL I would expect underestimating the uncertainty to result in poorer performance for the deep ensembles. Can the authors comment on why they think this is occurring? Is it the case that NTKGP consistently overestimates the uncertainty near the data, leading to poor(er) predictive performance when there is not dataset drift? This seems counter-intuitive given that the ensembles can be interpreted as valid approximations to the true Bayesian posterior predictive, and does not appear to be an issue in either Figure 1 or Figure 3 (left). I believe this point deserves more discussion than the remark in Lines 275-277.

Correctness: Unfortunately, I am not familiar enough with the NTK literature to assess the correctness of the derivations. I followed the derivations and proofs to the best of my ability, and did not find any glaring flaws.

Clarity: I found the paper to be very clear, and enjoyed reading it very much. The motivation is well-argued and discussed, and the logical progression of the paper is very easy to follow.

Relation to Prior Work: The authors also do a good job of discussing previous work, and how the current paper fits into the context of related work. The authors make clear use of the recent literature to motivate their work and provide the necessary background for their proposed development. Further, I found section 3.2 to be particularly useful in understanding the differences between the most closely related methods. In particular, Table 1 provides a very concise comparison of the limiting distributions of the different methods.

Reproducibility: Yes

Additional Feedback: - Toy-data experiment: The supplementary materials mention that $K=20$ ensemble members were used in this experiment, which is surprisingly few. It may also be useful to discuss the effect of the number of ensemble members on the resulting predictive distribution, and perhaps provide an empirical demonstration of this effect (this can be in the supplementary material if space is a constraint). For example, it would be interesting to see the results of this experiment on for each method when varying $K$. - If possible, I think it would be very useful to bring Fig 4 up from the supplementary material into the main text. It provides a lot of intuition Typos: - Line 173: $\mSigma (\vx, \vx')$, the first bracket seems to be subscripted unnecessarily. - Line 533: 'an library' -> 'a library' Post-discussion update: ---------------------------- I have reviewed the authors' responses and the additional reviews. I am still quite strongly in favour of accepting the paper based on its contributions. I disagree with some of the stated concerns regarding the technical / conceptual contributions of the paper, as well as with concerns regarding it practical applicability. 1. Conceptual contributions: several papers have recently shown that, on a suite of tasks regarding calibrated uncertainty, ensemble based methods tend to consistently outperform more principled approximations to posterior inference. It has also been claimed that ensemble methods can be viewed as principled approximations. The central contribution of this paper, as I see it, is to address and make steps towards resolving these questions. Namely, using the NTK limiting behaviour we can see precisely that (1) standard ensembles cannot be interpreted as principled approximations to the posterior, and (2) there is a "simple" modification to the training procedure such that ensemble members _can_ be interpreted as exact samples from the posterior (in the infinite width limit). The resolution of this question is an important contribution. The provided propositions, whether straightforward or not, are useful in validating the claims made by the authors. 2. Practical applications: the authors provide several experiments that demonstrate the usefulness of the proposed modification in relevant tasks and settings. More so when considering the additional experiments provided in the rebuttal. I am less worried about standard "large-scale" classification experiments. The authors (correctly, in my opinion) focus instead on several experiments where calibrated uncertainty is of particular importance (such as the o.o.d. tasks), which are better suited to evaluate the properties of the proposed method. I agree that scalability is a question that should be addressed, but on further reading of the paper and the authors' rebuttal, I am less concerned about this as the computation overhead is "negligible compared to standard deep ensembles...". As such, I am maintaining my score, and endorse the acceptance of the paper.


Review 3

Summary and Contributions: Post-rebuttal update: The new experiment using Myrtle CNN is nice to have, and suggests that the proposed method could be useful in practical applications. I have upgraded my score from 4 to 5 for this reason. However, I still don't think the manuscript is ready for publication, mainly because it's unclear whether the proposed method could faithfully recover the GP posterior as advertised, in real-world scenarios (complex network architecture and adam optimizer instead of SGD). My main concerns were that (1) in practice, for commonly-used NN architectures the kernel regime will end very quickly and (2) if we are in the kernel regime, the fast decay of the eigenspectrum means that convergence to the true GP posterior is prohibitively slow. The authors responded that (2) can be alleviated by the regularization effect of the observation noise parameter, but from the probabilistic modeling perspective, observation noise should be learnt to maximize likelihood and not be treated as a regularization parameter and tuned as such, and in many problems (e.g. classification with high-quality labels) its true value can be small, so the regularization effect wouldn't be significant. Another question I have from a new round of reading is that, it seems that for the Bayesian interpretation to hold, the noise parameter should be fixed throughout the training. If this is true, it would further limit the practical utility of the proposed method, if we are really interested in posterior inference for GP (since in most cases we need to learn the noise parameter alongside the inference process). I think for the manuscript to be ready for publication, there should be ablations comparing the learnt predictive distribution and the true GP posterior on slightly larger-scale problems: as R4 have noted, computing the NTK for a 3-layer ConvNet on MNIST should have been doable on a single GPU. There should also be further discussion on the aforementioned issues. ====== This work proposes a modified ensemble algorithm that converges to the Gaussian process posterior (for normal likelihood), assuming the neural tangent kernel is invariant throughout training. In two setups, the algorithm is shown to improve uncertainty quantification on OOD data.

Strengths: The modification to standard ensemble training is novel, and empirical improvement is demonstrated.

Weaknesses: * Convergence to GP posterior in the kernel regime is not very characteristic of real-world scenarios, since (1) there is substantial evidence showing that in practice, with finite-width NNs and different optimizers from vanilla GD, the NTK will not stay constant during training, and the change of NTK is crucial for the empirical success of DNNs (see e.g., Chizat, Oyallon and Bach, 2019; Kopitkov and Indelman, 2019; Woodworth et al, 2020; Lee et al 2020), and (2) if we are for some reason in the kernel regime, convergence takes a prohibitively long period of time due to the fast decay of eigenvalues, and may very well offset the benefit of avoiding taking N^3 time to compute the exact GP posterior. * For the above reason, I strongly believe that there should be a thorough empirical evaluation to demonstrate the efficacy of this approach. Currently non-synthetic experiments are only performed on two datasets (Airline and MNIST-vs-notMNIST), using feed-forward networks, while recent work on uncertainty estimation typically adapts more complex network architectures and tests on a wider range of datasets, see, e.g. (Maddox et al, 2019; Ovadia et al, 2019). References (updated after rebuttal): * Chizat, Lenaic, Edouard Oyallon, and Francis Bach, 2020, On lazy training in differentiable programming. * Dmitry Kopitkov, Vadim Indelman, 2019, Neural Spectrum Alignment: Empirical Study. * Blake Woodworth et al, 2020, Kernel and Rich Regimes in Overparametrized Models. * Lee et al, 2020, Finite Versus Infinite Neural Networks: an Empirical Study. * Maddox et al, 2019, A Simple Baseline for Bayesian Uncertainty in Deep Learning. * Ovadia et al, 2019, Can You Trust Your Model’s Uncertainty? Evaluating Predictive Uncertainty Under Dataset Shift

Correctness: Yes.

Clarity: Yes.

Relation to Prior Work: Yes.

Reproducibility: Yes

Additional Feedback: For improvements, I think it is most important to expand the evaluations and test with a wider range of datasets *and* network architectures. The settings in Ovadia et al (2019) or Maddox et al (2019) can be followed; alternatively, it would also be convincing to experiment with one or two new tabular datasets, and test under a setup similar to Sec 4.2 in Ovadia et al (2019).


Review 4

Summary and Contributions: This paper devises a method of constructing and training finite neural network approximations to Gaussian processes with neural tangent kernels. The key portions to the work are 1) incorporating a JVP into the prediction of the network and 2) ensembling over several different independently trained models. In the infinite limit, the finite neural network + the JVP is shown to have the same distribution as the equivalent GP. Experiments are shown that the ensembled models typically outperform standard deep ensembles on regression and small-ish classification tasks. I'd really like to see more comparisons to the (analytic) NTK GP and more discussions/experiments of the differences between the parameter and function space regularization schemes. {{Post-rebuttal updates to reviews are shown in brackets. I'm keeping my score at a weak accept; however, I commend the authors for their further experiments and clarifications. Currently, R3 makes several good comments that should be addressed in the camera ready.}}

Strengths: Significance: To my understanding, there has been a lot of research recently on figuring out how to practically use the neural tangent kernel for better uncertainty quantification in deep learning. This work seems to be one of the first to be able to successfully apply these theoretical advances to practical use cases, and especially just beyond pure predictive accuracy [31,34]. Relevance: The intersection of Gaussian processes, uncertainty quantification in deep learning, and the theoretical understanding of deep networks is clearly at the core of the NeurIPS community. This work is simultaneously theoretical and practical -- I hope the authors eventually integrate this into Jax and/or Jax's Neural Tangents library. Novelty: The authors manage to get around one of the weaknesses of work such as [34] by exploiting the prior distribution of the randomly initialized neural network. As such, they have a trainable neural network architecture that converges to an equivalent Gaussian process in the infinite limit, which to my knowledge is pretty much novel for architectures beyond one layer. The related NNGP [15-19] is about _untrained_ networks while the NTK gradient flow dynamics are not quite real GPs and are from initialization anyways. Empirical Evaluation: The experiments performed take this well beyond a theory paper. The out of distribution experiments are supposed to demonstrate that why we should care about the NTK GP approximation and to show that it will eventually be practically useful and could serve as a Bayesian version of deep ensembles. Theoretical grounding: Overall, the theoretical grounding is pretty clear, and the proof writing is quite well done when necessary. The method comes across as a theoretical union of works on infinite neural networks and deep ensembling.

Weaknesses: Relevance: Nothing significant. Novelty: Nothing major, it's a clever idea to train an ensemble of neural networks that has the asymptotic width distribution as an analytic NTK GP. In many ways however, it is practically a straightforward extension of [22, 23] by just adding the Jacobian term into the network. Theoretical grounding: I am a little concerned that in the infinite limit you are just training a NTK GP, so why not just use the NTK GP kernel method in most of the small-medium sized data situations in the paper. Obviously, scalability is a concern (the airlines experiment is certainly not easy for an exact kernel method) and not all architectures are analytically tractable. However, it feels like the model will ultimately inherit both the good and bad performance of an analytic NTK GP, which isn't that difficult to use in Jax. So, when possible, why not just use a NTK GP by computing the kernel in Jax as you get the predictive uncertainties for free from the gp representation, rather than having to train many neural networks. Experimental Evaluation: Beyond just comparing to [23] when possible, why not also compare to the analytic NTK GP on the MNIST experiment? There's no kernel learning involved here, so you just need to form and solve the kernel inverse problem, which is possible on a single GPU using the neural tangents library with a little bit of engineering effort [31]. It'd be interesting to see how well the approximation method performs in comparison to the true GP here. I find the lack of a real detailed comparison to the analytic NTK GP to be the weakest part of the experiment section and would certainly like to see the comparison in the rebuttal. {{Thank you for the comparison on the two moons problem; it's interesting to see that the sin activation function strongly outperforms the ReLU GPs here. However, I do think it's entirely possible to compute the analytic NTK gp on the MNIST experiment with a single GPU in a couple of hours -- you'd probably have to do a little bit of wrapping the kernel using `nt.utils.batch.__serial` if I remember correctly.}} Overall, it seems like the results are somewhat of a wash between the parameter and function space regularization techniques.. It seems like NTKGP-fn outperforms NTKGP-param in both panels of Figure 2 (lower NLL on all but the smallest training set and lower RMSE at most confidence thresholds) on the right panel of Figure 3 (Figure 2 is a wash for all methods it seems like). A couple of questions result: - Why not just present NTKGP-fn as the primary method instead of NTKGP-param? Is it simply an interpretation difference? - Show NTKGP-fn in Figure 1 if possible - I'm curious to know if it performs well in this sanity check. {{These would have been nice to see.}} Given that the model decomposes into the forwards pass and the Jacobian term, would it be possible to show how the prediction decomposes into these two terms, as well as the predictive uncertainty? {{Thanks for the clarification; however, seeing those terms in a plot would have been nice to disentangle what is giving the good predicitive uncertainties.}} Significance: While the method proposed seems to work reasonably well, I'm not entirely sure how it's more than a little bit better in practice than deep ensembles. Is the benefit of Bayesian-ness the only real benefit here? If so, then why not show an application (active learning?) where the Bayesian-ness is actually really helpful... As it stands, this feels like an easy trick to make an ensemble Bayesian, much like [22,23], but performance wise doesn't go much beyond [23] on the regression problem. It's also slower on the forwards pass than Bayesian style methods like [22,23] due to the forced inclusion of the Jacobian term. {{Thanks for the updates on how this method would be used in practice. Again, I find the discussion of its usefulness in practice quite fair and balanced overall.}}

Correctness: Claims and method: I gave a reasonably detailed reading to the proof of Proposition 1/(Appendix it's 3) and it looks fine to me. The discussion of the extra memory and computation costs of the proposed (and comparison) methods in Appendix G is quite fair overall. Line 111: Why should \Theta^{\leq L} be PD? I know that in proposition 4 you showed that the predictive covariances are ordered in this manner, but it's not immediately obvious to me why \Theta \geq K. {{Thanks for the clarification.}} Empirical methodology: Figure 1: I am a little concerned that the far right doesn't quite match as well as would be expected. A priori, it's probably fine that given a finite number of samples, the predictive covariance doesn't match, but the means in the region of interpolation between the green data points is a good deal different. Is this attributable to the finite samples or to the finite widths? Perhaps the authors could ablate this point by varying the width of the network and the number of samples used in the ensemble here. Similarly, given the obvious resemblance to some other anchored ensembles, why no comparison to [23] in the airline experiments? It looks like that method is about as scalable as the other deep ensembles methods, while the performance in [23] definitely outperforms deep ensembles in some aspects. See above box for more descripton on empirical methodology.

Clarity: Overall, the paper is very easy to read and enjoyable in some parts even. That being said, I have some minor and easily fixable concerns about experimental presentation: Line 152: You need to be a bit more careful to distinguish your approach from the kernel method that uses a NTK as a kernel and then computes a GP around that - e.g. GP(0, NTK). That could more properly be called a NTK-GP, rather than the approximate NTK GP that you've designed a loss to train towards the analytic NTK GP. Throughout the font legends and point sizes on the plots is considerably too small and they need to be zoomed in a bit. I reviewed this paper on a 13" laptop and used a standard PDF reader, and had to zoom into make each plot fill nearly the entire screen to really be able to interpret the plot. Figure 2 left: Those lines could probably be connected but regardless, the size of the points makes it very difficult to read and compare. Figure 2 right: The y label is almost impossible to read here. These two plots could be connected and use a joint legend (e.g. matplotlib subplots). A similar point appears for Figure 3 as well. Figure 1: It's probably not great practice to make the data points bright green and the underlying shading red/blue. e.g designing for colorblindness. The true function is also confusing as all of the models are just poor fits for the data; a slightly less steep true function would probably resolve that issue. Figure 4: the bottom row in the appendix could definitely use some distinguishing between ensembles of 5 and 10.

Relation to Prior Work: A more principled alternative to label smoothing is given in the context of approximating a Dirichlet likelihood, which then produces a heteroscedastic GP in their context (https://papers.nips.cc/paper/7840-dirichlet-based-gaussian-processes-for-large-scale-calibrated-classification.pdf). However, it could be easily used as a more straightforward way of converting classification into regression. Comparison to other Bayesian ensembling approaches would be useful -- for example [23] as well as the MultiSwag method of [14] if possible. However, I think that the proposed method is pretty clearly different from those approaches.

Reproducibility: Yes

Additional Feedback: Appendix line 434: you should probably mention that this is Thm. 29.4 in [42]. Eq. 23 is fine but it'd be nice to explicitly write out what this expectation converges to rather than wrapping everything in the result in Eq 24. Footnote 1: The h.c. marker is somewhat confusing throughout as it takes the reader a little bit of time to realize what matrix is having its hermitian conjugate taken. The excessive generality here is also a little confusing, as "transpose" would probably also fit perfectly fine given that real inputs are almost always used for kernel methods. Line 533: It's not just forward-mode AD that has the capability to do JVPs. It's native (although a bit tricky) to do in reverse-mode AD software such as PyTorch (https://pytorch.org/docs/stable/autograd.html#functional-higher-level-api although the capability has been around for a while).

[Author Response · NeurIPS 2020]

We thank reviewers **R1**, **R2**, **R3**, **R4** for their time and constructive reviews on our submission, which we will incorporate
to improve our paper. Due to limited space, we will only be able to address the major points from the reviews:

**Benefits of a GP posterior ensemble interpretation** (addressed to **R1** & **R4**) We agree with **R2** that a key strength of
our work is that it "provides a formal treatment of the relationship between deep ensembles and Bayesian posterior
predictive distributions". Posterior inference offers a principled way to convert prior beliefs into predictive uncertainties,
and provides o.o.d. robustness via Bayesian marginalisation [14]. Moreover, Bayesian ML has a rich history [3] and is
an active research frontier. One practical benefit to the GP posterior interpretation is selecting hyperparameters, like
activation, of the NTK/NN architecture (akin to choosing GP kernel) that best model prior beliefs about data. For
example, the NNGP/NTK correspondence allows one to deduce that Sine activation can alleviate overconfidence (see
[43]) of ReLU deep ensembles on Two Moons classification. This is because the ReLU kernels do not decay away
from the training data, as can be seen in Eqns S14,15 of [21], unlike the Sine kernels, as can be seen in `stax.py` of
the Neural Tangents library [31]. In the left 3 plots below, we demonstrate this empirically (with two layer NNs of
width 500, MSE trained with scaled one-hot regression targets and no observation noise, which are then fed into cross
entropy to get probabilities): we see that both deep ensembles [11] and NTKGP analytic (with small noise $\sigma^2 > 0$ added
for numerical stability, for **R4**) are overconfident with ReLU activation (denoted by blue and red shaded regions), but
NTKGP-param with Sine activation has low confidence (white regions) away from the training data (points), as desired.

**Additional experiment on CIFAR-10** (**R2**, **R3**) We repeated the MNIST/NotMNIST experimental setup using the
Myrtle-10 CNN with 100 channel width (Shankar et al., arXiv:2003.02237) trained on CIFAR-10 with SVHN o.o.d.
test set. We changed our classification methodology to use MSE loss (using scaled one-hot regression targets with scale
selected via moment-matching with NTK, small $\sigma^2 > 0$) before temperature scaling on a validation set. In the right 2
plots above, we see that our NTKGP methods perform slightly worse on in-distribution test accuracy (<1% higher error),
but outperform all baselines on o.o.d. detection in the Error vs Confidence plot (far right). For instance, NTKGP-fn
exceeds baselines by between 8-10% accuracy on (combined in-dist+o.o.d.) test points with confident predictions (e.g.
confidence $\tau = 0.8$). This o.o.d. performance gain is crucial for safety-critical applications (e.g. self-driving cars).
**Computational overhead** (**R1**, **R2**, **R4**) We would like to clarify that for a training set of fixed-size (e.g. CIFAR-10)
the train-time overhead of our methods is negligible compared to standard deep ensembles [11]: one can obtain and
store our fixed additive JVPs $\delta$ in a single pass over the training data. This was mentioned on lines 528-531. For
test-time constrained applications, we could apply distillation techniques, as is common with standard deep ensembles.
**Novelty** (**R1**, **R4**) Though presentation has been simplified, we respectfully disagree that the novelty of our paper is
straightforward. We are (to our knowledge) the first to consider GP(0,NTK) prior instead of GP(0,NNGP), in order
to align posterior inference with optimisation of *all* NN layers. Our contributions are distinct to, not extensions of,
[22, 23], and give different limiting predictive distributions to [22], see Table 1. Also, if there is no modelling of
observation noise, $\sigma^2 = 0$, then RP-param [22, Eq. 4] and anchored ensembles with MSE [23, Eq. 8] become standard
deep ensembles [11]. On the other hand, NTKGP-param still retains its posterior interpretation (Corollary 1), using
fixed additive JVP corrections with no regularisation nor noisy targets. This is the case in the two moons ensembles
above. It is only when modelling observation noise that we synergise our methods with [22].
**Individual responses** (**R1**) We contest the "marginal" empirical improvement of our work: Figure 3 (right) depicts
significant gains of our methods for o.o.d. NotMNIST detection over baselines. The NTK & standard parameterisations
are introduced in Appendix A. (**R2**) We believe the slightly worse in-distribution test performance of our methods
can be alleviated with thorough NTK hyperparameter tuning. (**R3**) When modelling observation noise, $\sigma^2 > 0$, our
regularisation scheme (Appendix D) enables closer alignment to the kernel regime in standard parameterisation (lines
492-499) and nullifies problems caused by the fast decay of NTK eigenvalues (lines 84-86). (**R4**) $\Theta \succeq \mathcal{K}$ follows from the
NTK being a sum of p.d. contributions from different layers and $\mathcal{K}$ is the contribution from last layer, see Eq. S29 of [21].
JVPs are more memory-efficient in forward-mode than reverse-mode AD; we will add this. Please see lines 511-513
for discussion of parameter and function space methods; our code is open-source and we are working with the Neural
Tangents authors [31] to integrate our work. It is unclear if analytic NTKGPs are preferable to analytic NNGPs when
both are tuned, due to cost and predictive-mean performance (see §3.2 of Lee et al., arXiv:2007.15801); we focus on
giving a posterior interpretation to deep ensembles for wide but finite NNs, and lack the compute needed for comparisons
of large-scale analytic NTKGP/NNGPs. For the prediction decomposition, setting $\Theta_{\mathcal{X}\mathcal{X}}^{\sigma} = \Theta(\mathcal{X}, \mathcal{X}) + \sigma^2 I$, we obtain:

50
$$\tilde{f}_\infty(\boldsymbol{x}^*) = \Theta(\boldsymbol{x}^*, \mathcal{X})\big[\Theta_{\mathcal{X}\mathcal{X}}^{\sigma}\big]^{-1}\mathcal{Y} + \underbrace{f_0(\boldsymbol{x}^*) - \Theta(\boldsymbol{x}^*, \mathcal{X})\big[\Theta_{\mathcal{X}\mathcal{X}}^{\sigma}\big]^{-1}f_0(\mathcal{X})}_{f_0} + \underbrace{\delta(\boldsymbol{x}^*) - \Theta(\boldsymbol{x}^*, \mathcal{X})\big[\Theta_{\mathcal{X}\mathcal{X}}^{\sigma}\big]^{-1}\delta(\mathcal{X})}_{\delta}$$

[Meta-Review · NeurIPS 2020]

*PROS: provides an understanding the relationship between deep ensembles and posterior inference, several experiments that demonstrate the usefulness of the proposed method, including additional ones in the rebuttal. *CONS: high computational overhead. R3 makes several good comments that should be addressed in the final version Meta-reviewre recommendations: The paper is boerderline but I recommend acceptance for the reasons mentioned by R2 in his post rebuttal update. In particular, that this work presents a formal argument, based on recent work on neural tangent kernel (NTK), that standard deep ensembles do not have valid interpretations as Bayesian posterior predictive approximations. I recommend the authors to take into account the reviewers points to improve the paper for the final version.